# FORMALIZING SPURIOUSNESS OF BIASED DATASETS USING PARTIAL INFORMATION DECOMPOSITION

## ABSTRACT

Spuriousness arises when there is an association between two or more variables in a dataset that are not causally related. Left unchecked, they can mislead a machine learning model into using the undesirable "spurious" features in decision-making over the "core" features, hindering generalization. In this work, we propose a novel explainability framework to disentangle the nature of such spurious associations, i.e., how the information about a target variable is distributed among the spurious and core features. Our framework leverages a body of work in information theory called Partial Information Decomposition (PID) to first decompose the total information about the target into four non-negative quantities namely *unique information (in core and spurious features respectively), redundant information, and synergistic information.* Next, we leverage this decomposition to propose a novel measure of the spuriousness of a dataset that steers models into choosing the spurious features *over* the core. We arrive at this measure systematically by examining several candidate measures, and demonstrating what they capture and miss through intuitive canonical examples and counterexamples. Our proposed explainability framework *Spurious Disentangler* consists of segmentation, dimensionality reduction, and estimation modules, with capabilities to specifically handle high dimensional image data efficiently. Finally, we also conduct empirical evaluation to demonstrate the trends of unique, redundant, and synergistic information, as well as our proposed spuriousness measure across several benchmark datasets under various settings. Interestingly, we observe a novel tradeoff between our measure of dataset spuriousness and empirical model generalization metrics such as worst-group accuracy, further supporting our proposition.

## 1 INTRODUCTION

While machine learning is rapidly percolating into almost every aspect of our lives, its success is heavily determined by the datasets used for training or fine-tuning. Spurious patterns (Haig, 2003) arise when two or more variables are associated in a dataset even though they do not have a causal relation. For example, image classifiers trained on the Waterbird dataset (Wah et al., 2011) learn to use the background rather than the foreground (actual characteristics of the bird) for classification, because most waterbirds are photographed on a water background (see Fig. 1). This pattern in the dataset misleads a machine learning classifier into learning an undesirable spurious link between the target label (bird type) and background ("spurious" feature) as opposed to the foreground (core feature). Spuriousness in datasets may result in deceptively high performance on in-distribution datasets but significantly hinders generalization on out-of-distribution datasets, e.g., accuracy on minority groups like waterbirds with land background is low (Lynch et al., 2023; Sagawa et al., 2019; Puli et al., 2023).

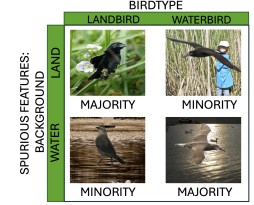

Figure 1: Spurious patterns due to sampling bias.

Despite advances in dataset-based and model-training-based approaches to mitigate such spurious patterns (Kirichenko et al., 2022; Izmailov et al., 2022; Wu et al., 2023; Ye et al., 2023; Liu et al., 2023), this notion of spuriousness in any given dataset has classically lacked a formal definition. To address this gap, in this work, we ask the following question: *Given a dataset and a split of core and spurious features, how do we quantify the undesirable spuriousness of the dataset which steers machine learning models into choosing the spurious features over the core features?*

Towards answering this question, we present an information-theoretic explainability framework to disentangle the nature of such spurious associations, i.e., how the information about the target variable is distributed among the spurious and core features. To this end, we leverage a body of work in information theory called Partial Information Decomposition (PID) (Bertschinger et al., 2014; Banerjee et al., 2018), which has its roots in statistical decision theory. We note that classical information-theoretic measures such as mutual information (Cover & Thomas, 2012) captures the entire statistical dependency between two random variables but fail to capture how this dependency is distributed among those variables, i.e., the structure of the multivariate information. Partial Information Decomposition (PID) addresses this nuanced issue by providing a formal way of *disentangling* the joint information content between the core and spurious features into non-negative terms, namely, *unique, redundant, or synergistic information* (see in Section 2). We leverage this decomposition to systematically arrive at a novel measure of dataset spuriousness with empirical evaluation on high-dimensional image datasets. This work provides a more nuanced understanding of the interplay between spurious and core features in a dataset that can better inform dataset quality assessment.

Our main contributions can be summarized as follows:

**Unraveling nature of spurious associations leveraging Partial Information Decomposition:** Novel to this work, we investigate the problem of learning spurious patterns from a dataset through the lens of partial information decomposition (PID). We leverage PID to disentangle the total information about a target ($Y$) in the core ($F$) and spurious ($B$) features into four non-negative terms: *unique information (in core and spurious features respectively), redundant information, and synergistic information* (see Proposition 1). We elucidate four types of statistical dependencies captured by these PID terms (see Fig. 3), providing pre-emptive insights on when an optimal classifier might find a spurious feature more informative or useful than the core features. We establish how unique information quantifies the informativeness of a random variable over another for predicting $Y$ (see Theorem 1 for interpretability insights, also leveraging Blackwell Sufficiency). Then, redundant information turns out to be the common information that can be obtained from either the spurious or core features, allowing a predictor to potentially choose either without a preference. An interesting term is the synergistic information that captures scenarios when both spurious and core features are jointly informative about the target $Y$ but not individually.

**Novel information-theoretic measure of spuriousness:** Though many works attempt to prevent a model from learning spurious patterns, there is limited theoretical understanding of how to quantify the spuriousness of a dataset, given a choice of core and spurious features. In this work, we leverage PID to propose a novel measure of the undesirable spuriousness of a dataset ($M_{sp}$) that steers predictors into choosing the spurious features over the core (see Proposition 2). We arrive at this measure systematically by examining several candidate measures, and demonstrating what they capture and miss through intuitive canonical examples and counterexamples. Our measure provides a fundamental understanding of which features can be more informative for a classification task, paving a pathway for dataset quality assessment and interpretability.

**Spuriousness Disentangler: An autoencoder-based explainability framework:** We propose an autoencoder-based explainability framework that we call – Spuriousness Disentangler – to compute the PID values and our spuriousness measure for high dimensional image data. The framework consists of three modules: (i) Segmentation: If desired, our framework performs segmentation to separate the foreground (core features $F$) and background (spurious features $B$) for every image; (ii) Dimensionality Reduction: An autoencoder converts high-dimensional images into lower-dimensional, discrete feature representations. Along the lines of Guo et al. (2017), the dimensionality reduction and clustering are efficiently performed through minimization of a joint loss function. We also incorporate a bottleneck structure from Sadeghi & Armanfard (2023) to have a more informative lower dimensional representation; (iii) Estimation: The final step includes the estimation of the joint probability distribution of the acquired lower-dimensional representation followed by computing PID values and our measure $M_{sp}$. The computation is performed by solving a convex optimization problem using the Discrete Information Theory (DIT) package (James et al., 2018).

**Empirical results:** Since our proposed framework is a pre-emptive dataset explainability framework, the goal of our experiments is to show broad agreement between our anticipations from the dataset before training and the post-training behavior of models for various experimental setups. We observe a negative correlation between our proposed measure of dataset spuriousness $M_{sp}$ and post-training model generalization metrics, such as the worst-group accuracy. We also study Grad-CAM (Selvaraju

et al., 2017) visualizations and intersection-over-union (IoU) metric (Rezatofighi et al., 2019) to further confirm which features are actually being emphasized by the model.

**Related Works:** There are several perspectives on spurious correlation (see Haig (2003); Kirichenko et al. (2022); Izmailov et al. (2022); Wu et al. (2023); Ye et al. (2023); Liu et al. (2023); Stromberg et al. (2024); Singla & Feizi (2021); Moayeri et al. (2023); Lynch et al. (2023) and the references therein; also see surveys Ye et al. (2024); Srivastava (2023); Ghouse et al. (2024)). Spuriousness mitigation techniques are broadly divided into two groups: (i) Dataset-based techniques (Goel et al., 2020; Kirichenko et al., 2022; Wu et al., 2023; Moayeri et al., 2023; Liu et al., 2021) and (ii) Learning-based techniques (Liu et al., 2023; Yang et al., 2023; Ye et al., 2023; Zhang et al., 2022). Among dataset-based techniques, Kirichenko et al. (2022) shows that last-layer fine-tuning of a pre-trained model with a group-balanced subset of data is sufficient to mitigate spurious correlation. Wu et al. (2023) proposes a concept-aware spurious correlation mitigation technique. A recent work (Wang & Wang, 2024) looks into the problem of spurious correlations through the mathematical lens of separability of the spurious and core features under mixture of Gaussian assumptions (also assuming a split between core and spurious). Ye et al. (2023) discusses how the noise in the core feature plays a role in a model's reliance on it. Our novelty lies in investigating the problem of spurious patterns through the lens of Partial Information Decomposition, rooted in statistical decision theory, focusing on quantifying the spuriousness of a dataset for interpretability and quality assessment. Our work isolates four specific types of statistical dependencies in the dataset, providing a more nuanced understanding (see Fig. 3) going beyond identifying a model's reliance on a specific feature.

Partial Information Decomposition (Williams & Beer, 2010; Bertschinger et al., 2014) is an active area of research. PID measures are beginning to be used in different domains of neuroscience and machine learning (Tax et al., 2017; Dutta et al., 2021; Hamman & Dutta, 2024; Ehrlich et al., 2022; Liang et al., 2024; Wollstadt et al., 2023; Mohamadi et al., 2023; Venkatesh et al., 2024). However, interpreting spuriousness in datasets through the lens of PID and observing novel empirical tradeoffs between spuriousness and worst-group accuracy is unexplored. Additionally, there is limited work on calculating PID values for high dimensional multivariate continuous data. Some existing works (Dutta et al., 2021; Venkatesh et al., 2024) handle continuous data with Gaussian assumptions while (Pakman et al., 2021) considers one-dimensional multivariate case. Hence, estimating PID for high-dimensional data through proper dimensionality reduction and discretization is also fairly open. For dimensionality reduction, different learning based methods exist (Hotelling, 1933; Law & Jain, 2006; Lee & Verleysen, 2005; Wang et al., 2015; 2014; Sadeghi & Armanfard, 2023). Similarly, for discretization, different clustering algorithms exist, e.g., k-means clustering (MacQueen et al., 1967; Bradley et al., 2000), deep embedded clustering (Xie et al., 2016). There are also some works that try to separate spurious and core features in the feature space of deep neural networks using external feedback (Sohoni et al., 2020; Kattakinda et al., 2022). In this work, along the lines of an autoencoder-based clustering setup in Guo et al. (2017), we train an autoencoder to jointly learn a good lower-dimensional representation of the input image data in a self-supervised manner (with additional bottleneck structure from Sadeghi & Armanfard (2023)) while also clustering simultaneously to deal with the challenge of high dimensional and continuous image data.

## 2 PRELIMINARIES

Let $X = (X_1, X_2, \ldots, X_d)$ be the random variable denoting the input (e.g., an image) where each $X_i \in \mathcal{X}$ which denotes a finite set of values that each feature can take. The core features (e.g., the foreground) will be denoted by $F \subseteq X$, and the spurious features (e.g., the background) will be denoted by $B = X \backslash F$. We typically use the notation $\mathcal{B}$ and $\mathcal{F}$ to denote the range of values for the spurious and core features. Let $Y$ denote the target random variable, e.g., the true labels which lie in the set $\mathcal{Y}$, and the model predictions are given by $\hat{Y} = f_\theta(X)$ (parameterized by $\theta$). Generally, we use the notation $P_A$ to denote the distribution of random variable $A$, and $P_{A|B}$ to denote the conditional distribution of random variable $A$ conditioned on $B$. Depending on the context, we also use more than one random variable as sub-script, e.g., $P_{ABY}$ denotes the joint distribution of $(A, B, Y)$. Whenever necessary, we also use the notation $Q_A$ to denote an alternate distribution on the random variable $A$ that is different from $P_A$. We also use the notation $P_{A|B} \circ P_{B|C}$ to denote a composition of two conditional distributions given by: $P_{A|B} \circ P_{B|C}(a|c) = \sum_{b \in \mathcal{B}} P_{A|B}(a|b) P_{B|C}(b|c) \ \forall a \in \mathcal{A}, \ c \in \mathcal{C}$, where $\mathcal{A}, \mathcal{B}$ and $\mathcal{C}$ denote the range of values that can be taken by random variables $A$, $B$, and $C$.

**Background on Partial Information Decomposition:** We provide a brief background on PID that would be relevant for the rest of the paper. The classical information-theoretic quantification of the total information that two random variables $A$ and $B$ together hold about $Y$ is given by the mutual information $\mathrm{I}(Y; A, B)$ (see (Cover & Thomas, 2012) for a background on mutual information). Mutual information $\mathrm{I}(Y; A, B)$ is defined as the KL divergence (Cover & Thomas, 2012) between the joint distribution $P_{YAB}$ and the product of the marginal distributions $P_Y \otimes P_{AB}$ and would go to zero if and only if $(A, B)$ is independent of $Y$. *Intuitively, this mutual information captures the total predictive power about $Y$ that is present jointly in $(A, B)$ together, i.e., how well can one learn $Y$ from $(A, B)$ together.* However, $\mathrm{I}(Y; A, B)$ only captures the total information content about $Y$ jointly in $(A, B)$ and does not unravel anything about what is unique or shared between $A$ and $B$.

PID (Bertschinger et al., 2014; Banerjee et al., 2018) provides a mathematical framework that decomposes the total information content $\mathrm{I}(Y; A, B)$ into four non-negative terms (also see Fig. 2):

$$\mathrm{I}(Y; A, B) = \mathrm{Uni}(Y{:}B|A) + \mathrm{Uni}(Y{:}A|B) + \mathrm{Red}(Y{:}A, B) + \mathrm{Syn}(Y{:}A, B). \quad (1)$$

Here, $\mathrm{Uni}(Y{:}A|B)$ denotes the *unique information* about $Y$ that is only in $A$ but not in $B$ and $\mathrm{Uni}(Y{:}B|A)$ denotes the *unique information* about $Y$ that is only in $B$ but not in $A$. Next, $\mathrm{Red}(Y{:}A, B)$ denotes redundant information (common knowledge) about $Y$ in both $A$ and $B$. Lastly, $\mathrm{Syn}(Y{:}A, B)$ is an interesting term that denotes the synergistic information that is present only jointly in $A, B$ but not in any one of them individually, e.g., a public and private key can jointly reveal information not in any one of them alone.

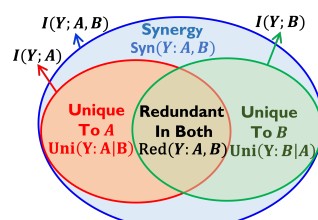

**Example to Understand PID.** Let $Z=(Z_1, Z_2, Z_3)$ with each $Z_i \sim$ i.i.d. Bern(1/2). Let $A = (Z_1, Z_2, Z_3 \oplus N)$, $B = (Z_2, N)$, and $N \sim$ Bern(1/2) which is independent of $Z$. Here, $\mathrm{I}(Z; A, B) = 3$ bits. The unique information about $Z$ that is contained only in $A$ and not in $B$ is effectively in $Z_1$, and is given by $\mathrm{Uni}(Z{:}A|B) = \mathrm{I}(Z; Z_1) = 1$ bit. The redundant information about $Z$ that is contained in both $A$ and $B$ is effectively in $Z_2$ and is given by $\mathrm{Red}(Z{:}A, B) = \mathrm{I}(Z; Z_2) = 1$ bit. Lastly, the synergistic information about $Z$ that is not contained in either $A$ or $B$ alone, but

Figure 2: $\mathrm{I}(Y; A, B)$ is decomposed into four non-negative terms: unique information in $A$, unique information in $B$, redundant information in both, and synergistic information in both.

is contained in both of them together is effectively in the tuple $(Z_3 \oplus N, N)$, and is given by $\mathrm{Syn}(Z{:}A, B) = \mathrm{I}(Z; (Z_3 \oplus N, N)) = 1$ bit. This accounts for the 3 bits in $\mathrm{I}(Z; A, B)$.

Defining any one of the PID terms suffices for obtaining the others. This is because of another relationship among the PID terms as follows (Bertschinger et al., 2014): $\mathrm{I}(Y; A) = \mathrm{Uni}(Y{:}A|B) + \mathrm{Red}(Y{:}A, B)$. Essentially $\mathrm{Red}(Y{:}A, B)$ is viewed as the sub-volume between $\mathrm{I}(Y; A)$ and $\mathrm{I}(Y; B)$ (see Fig. 2). Hence, $\mathrm{Red}(Y{:}A, B) = \mathrm{I}(Y; A) - \mathrm{Uni}(Y{:}A|B)$. Lastly, $\mathrm{Syn}(Y{:}A, B) = \mathrm{I}(Y; A, B) - \mathrm{Uni}(Y{:}A|B) - \mathrm{Uni}(Y{:}B|A) - \mathrm{Red}(Y{:}A, B)$ (can be obtained once both unique and redundant information has been obtained). Here, we include a popular definition of $\mathrm{Uni}(Y{:}A|B)$ from (Bertschinger et al., 2014) which is computable using convex optimization.

**Definition 1** (Unique Information (Bertschinger et al., 2014)). *Let $\Delta$ be the set of all joint distributions on $(Y, A, B)$ and $\Delta_P$ be the set of joint distributions with same marginals on $(Y, A)$ and $(Y, B)$ as the true distribution $P_{YAB}$, i.e., $\Delta_P = \{Q_{YAB} \in \Delta : Q_{YA} = P_{YA}$ and $Q_{YB} = P_{YB}\}$. Then,*

$$\mathrm{Uni}(Y{:}A|B) = \min_{Q \in \Delta_P} \mathrm{I}_Q(Y; A|B). \quad (2)$$

*Here $\mathrm{I}_Q(Y; A|B)$ denotes the conditional mutual information when $(Y, A, B)$ have joint distribution $Q_{YAB}$ instead of $P_{YAB}$.*

## 3 Main Results

### 3.1 Unraveling the Nature of Spurious Associations Leveraging PID

**Proposition 1** (Proposed Disentanglement). *For a given data distribution, the total predictive power of the spurious features $B$ and core features $F$ about the target variable $Y$ can be decomposed into four non-negative components as follows:*

$$\mathrm{I}(Y; F, B) = \mathrm{Uni}(Y{:}B|F) + \mathrm{Uni}(Y{:}F|B) + \mathrm{Red}(Y{:}F, B) + \mathrm{Syn}(Y{:}F, B). \quad (3)$$

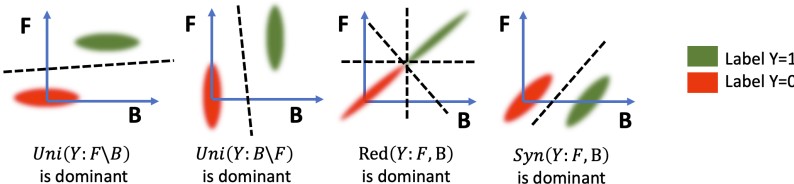

Figure 3: Canonical examples distilling four types of statistical dependencies involving core and spurious features when any one PID term is dominant and its effect on the Bayes optimal classifier. In the first two cases, unique information in either $F$ or $B$ is dominant, and they are indispensable to the optimal classifier. When redundant information is dominant, the optimal classifier can pick either $F$ or $B$ without preference. The fourth scenario is interesting where $B$ is independent of the label $Y$, and yet it contributes to the optimal classifier along with $F$.

For each term in Proposition 1, we now explain their nuanced role for any given dataset.

*Interpreting Unique Information* $\mathrm{Uni}(Y{:}B|F)$ *and* $\mathrm{Uni}(Y{:}F|B)$*:* Unique information captures information that is unique in one feature and cannot be obtained from another. To explain the role of unique information in interpreting spuriousness, we draw upon a concept in statistical decision theory called Blackwell Sufficiency (Blackwell, 1953) which investigates when a random variable is "more informative" (or "less noisy") than another for inference (also relates to stochastic degradation of channels (Venkatesh et al., 2023; Raginsky, 2011)). Let us first discuss this notion intuitively when trying to infer $Y$ using two random variables $F$ and $B$. Suppose, there exists a transformation on $F$ to give a new random variable $B'$ which is always equivalent to $B$ for predicting $Y$ (similar predictive power). We note that $B'$ and $B$ do not necessarily have to be the same since we only care about inferring $Y$. In fact, $B$ and $B'$ can have additional irrelevant information that do not pertain to $Y$, but solely for the purpose of inferring $Y$, they need to be equivalent. Then, $F$ will be regarded as "sufficient" with respect to $B$ for predicting $Y$ since $F$ can itself provide all the information that $B$ has about $Y$ (see Fig. 4 and first two cases of Fig. 3).

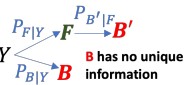

Figure 4: Blackwell Sufficiency

**Definition 2** (Blackwell Sufficiency (Blackwell, 1953)). *A conditional distribution $P_{F|Y}$ is Blackwell sufficient with respect to another conditional distribution $P_{B|Y}$ if and only if there exists a stochastic transformation (equivalently another conditional distribution $P_{B'|F}$ with both $B$ and $B' \in \mathcal{B}$) such that $P_{B'|F} \circ P_{F|Y} = P_{B|Y}$.*

In fact, the unique information $\mathrm{Uni}(Y{:}B|F)$ is 0 if and only if $P_{F|Y}$ is Blackwell sufficient with respect to $P_{B|Y}$ (see Theorem 1, the proof is given in the Appendix F).

**Theorem 1** (Interpretability Insights from Unique Information). *The following properties hold:*

- $\mathrm{Uni}(Y{:}B|F) \leq \mathrm{I}(Y;B)$ *and goes to 0 if the spurious feature $B$ is independent of the target $Y$. However,* $\mathrm{Uni}(Y{:}B|F)$ *may be 0 even if* $\mathrm{I}(Y;B) > 0$.

- $\mathrm{Uni}(Y{:}B|F) = 0$ ***if and only if*** $P_{F|Y}$ *is Blackwell sufficient with respect to* $P_{B|Y}$.

- $\mathrm{Uni}(Y{:}B|F) \leq \mathrm{Uni}(Y{:}B'|F')$*, i.e., it is non-decreasing if some features from the core set are moved to the spurious set, i.e., $B' = B \cup W$ and $F' = F\backslash W$.*

Since unique information $\mathrm{Uni}(Y{:}B|F) = 0$ if and only if $P_{F|Y}$ is Blackwell Sufficient with respect to $P_{B|Y}$, we note that $\mathrm{Uni}(Y{:}B|F) > 0$ captures the "departure" from Blackwell Sufficiency, and thus quantifies *relative informativeness. Intuitively, what this means is that for a data distribution, there is no such transformation on core feature $F$ that is equivalent to the spurious feature $B$ for the purpose of predicting $Y$. This essentially makes spurious feature $B$ indispensable for predicting $Y$, forcing a model to emphasize it in decision-making.* A similar argument can be made for $\mathrm{Uni}(Y{:}F|B)$. Furthermore, $\mathrm{Uni}(Y{:}B|F)$ also satisfies an intuitive property that as more features get categorized as spurious instead of core, the unique information in the spurious set would keep increasing.

*Interpreting Redundant Information* $\mathrm{Red}(Y{:}F, B)$*:* Redundant information about the target variable $Y$ is the information that can be obtained from either the spurious features $B$ or the core features $F$ without any preference towards either. We consider the following canonical example to interpret the role of redundant information $\mathrm{Red}(Y{:}F, B)$ for predicting the target variable $Y$ (third case of Fig. 3).

**Lemma 1** (Redundancy). *Let $B = Y + N_B, F = Y + N_F$ where noise $N_B$ and $N_F$ are Gaussian such that $N_B = N_F = N \sim \mathcal{N}(0, \sigma_N^2)$ and $N \perp\!\!\!\perp Y$. In this case, (i) an optimal predictor $\hat{Y}$ can either utilize $B$ or $F$ with neither being indispensable, i.e., $\hat{Y} = f(B)$ or $f(F)$ or $f(B, F)$; and (ii) $B$ and $F$ will only have redundant information with the other PID terms being $0$.*

*Interpreting Synergistic Information:* Synergistic information $\mathrm{Syn}(Y{:}F, B)$ is an interesting term that emerges when spurious features $B$ and core features $F$ together reveal more about the target variable $Y$ than what can be revealed by either of them alone. In essence, it is the "extra" or "emergent" information that arises only when multiple features interact, rather than when they are considered separately. Consider the example below to have an intuition on the role of this component.

**Lemma 2** (Synergy). *Let $B{=}N$, $F{=}Y + N$ where $Y \sim Bern(1/2)$, $N \sim \mathcal{N}(0, \sigma_N^2)$, $N \perp\!\!\!\perp Y$ and $\sigma_N^2 \gg 1$. Then, (i) an optimal predictor $\hat{Y} = f(F, B) = F - B$ (uses both $F$ and $B$); and (ii) $\mathrm{I}(Y; B)$ and $\mathrm{I}(Y; F) \approx 0$ but $\mathrm{I}(Y; B, F)$ is still significant due to synergistic information $\mathrm{Syn}(Y{:}B, F)$.*

For this example (fourth case in Fig. 3), both $F$ and $B$ alone will have limited predictive power when $N$ has high variance. However, using $F$ and $B$ together, one can perfectly predict $Y$, e.g., an optimal predictor is $\hat{Y} = f(F, B) = F - B$. Here $\mathrm{I}(Y; B) = 0$, and we also show that $\mathrm{I}(Y; F) \approx 0$ (see Lemma 8 in Appendix F). However, the synergistic information $\mathrm{Syn}(Y{:}F, B)$ is still significant. Since $\mathrm{I}(Y; F) \approx 0$, we contend that here $B$ essentially denoises the core feature $F$, enhancing its predictive power. Thus, synergistic information captures an interesting nuanced interplay between core and spurious, not captured by the other PID terms.

## 3.2 NOVEL INFORMATION-THEORETIC MEASURE OF SPURIOUSNESS

Our objective is to quantify a dataset's spuriousness which steers machine learning models towards the spurious features over the core features. To this end, we will examine some candidate measures ($M_{sp}$) of spuriousness through examples and counterexamples and systematically arrive at a measure that meets our requirements. Since we are trying to capture spuriousness which arises when the target variable $Y$ is associated with the spurious features $B$, we might first consider the mutual information $\mathrm{I}(Y; B)$ as a candidate measure for spuriousness since it captures the dependence between $Y$ and $B$.

**Candidate Measure 1.** $M_{sp} = \mathrm{I}(Y; B)$.

**Counterexample 1.** We refer to the example in Lemma 1 where $\mathrm{Uni}(Y{:}B|F) = 0$. Hence, $\mathrm{I}(Y; B) = \mathrm{Uni}(Y{:}B|F) + \mathrm{Red}(Y{:}F, B) = \mathrm{Red}(Y{:}F, B)$. Here, our candidate measure $M_{sp} = \mathrm{I}(Y; B)$ is positive which would indicate "spuriousness," i.e., undesirable steering towards $B$. However, in this case the model can use either spurious features $B$ or core features $F$ (see Lemma 1) without any preference. Thus, $\mathrm{I}(Y; B)$ is not well suited to be a measure of undesirable spuriousness.

Since redundant information can lead to the utilization of either spurious or core features, another candidate measure of spuriousness might be obtained by subtracting the desirable dependence $\mathrm{I}(Y; F)$ from the undesirable dependence $\mathrm{I}(Y; B)$, i.e., $M_{sp} = \mathrm{I}(Y; B) - \mathrm{I}(Y; F)$. For the example in Lemma 1, this new $M_{sp} = 0$, indicating no preference towards spurious or core features.

**Lemma 3.** *Let $B = Y + N_B, F = Y + N_F$ where noise $N_B$ and $N_F$ are standard Gaussian noises with $N_B \sim \mathcal{N}(0, \sigma_{N_B}^2)$, $N_F \sim \mathcal{N}(0, \sigma_{N_F}^2)$ and $N_B \perp\!\!\!\perp Y$, $N_F \perp\!\!\!\perp Y$. Now if $\sigma_{N_F}^2 \gg \sigma_{N_B}^2$, (i) the optimal classifier relies strongly on spurious feature $B$; and (ii) $\mathrm{Uni}(Y{:}B|F) > 0$.*

If $\sigma_{N_F}^2 \gg \sigma_{N_B}^2$, then $\mathrm{I}(Y; B) > \mathrm{I}(Y; F)$, i.e., $M_{sp} > 0$ (see Lemma 8 in Appendix F). In this case, the output of a model is more likely to be $\hat{Y} = f(B)$ and the model might be more prone to utilizing the spurious features $B$ (see Fig. 3). On the other hand, if $\sigma_{N_F}^2 \ll \sigma_{N_B}^2$, then $\mathrm{I}(Y; F) > \mathrm{I}(Y; B)$, i.e., $M_{sp} < 0$ . In this case, the output of the model is also more likely to be $\hat{Y} = f(F)$ and the model might lean towards the core features $F$. Hence, $M_{sp} = \mathrm{I}(Y; B) - \mathrm{I}(Y; F)$ might seem like a suitable measure to quantify spuriousness, i.e., steering models towards $B$ over $F$.

**Candidate Measure 2.** $M_{sp} = \mathrm{I}(Y; B) - \mathrm{I}(Y; F) = \mathrm{Uni}(Y{:}B|F) - \mathrm{Uni}(Y{:}F|B)$.

**Counterexample 2.** Consider Lemma 2 where the optimal predictor $\hat{Y} = F - B$ utilizing both the spurious features $B$ and core features $F$. Here, this $M_{sp} \approx 0$ (Lemma 2). However, for this particular example, since the prediction is jointly influenced by both core features $F$ and spurious features $B$,

Figure 5: Spuriousness Disentangler: An autoencoder-based explainability framework to handle high dimensional continuous image data with 3 modules: (i) Segmentation of images into background (spurious features) and foreground (core features); (ii) Dimensionality reduction involving an autoencoder with bottleneck and clustering; and (iii) Estimation of the joint distribution followed by the computation of PID values through convex optimization and computing $M_{sp}$.

we contend that a measure of spuriousness should not be $0$. The measure should therefore include a term that considers the joint contribution of both of these features, capturing the fact that here $B$ simply helps in denoising and enhancing the predictive capabilities of the core features $F$. This aspect is precisely captured by synergistic information $\mathrm{Syn}(Y{:}F, B)$. Hence, we also include it in $M_{sp}$, leading to the following proposed measure.

**Proposition 2** (Measure of Spuriousness $M_{sp}$). *Our proposed measure of spuriousness is given by:*

$$M_{sp} = \mathrm{Uni}(Y{:}B|F) - \mathrm{Uni}(Y{:}F|B) - \mathrm{Syn}(Y{:}F, B). \tag{4}$$

### 3.3 PROPOSED EXPLAINABILITY FRAMEWORK: SPURIOUSNESS DISENTANGLER

We propose an autoencoder-based explainability framework – that we call Spuriousness Disentangler – to disentangle the PID values and compute the measure $M_{sp}$ (see Fig. 5) for a given dataset. The framework mainly consists of three modules: segmentation, dimensionality reduction, and estimation.

**Segmentation:** The first step involves separating the foreground ($F$) from the background ($B$). For the Waterbird and CelebA datasets, publicly available segmentation masks ($m$) are utilized to achieve this separation, as illustrated in Fig.5. Since the Dominoes dataset is constructed synthetically by concatenating the foreground and background, we need not use segmentation mask for this dataset. For the Spawrious dataset, we generate masks using a pre-trained semantic segmentation model (see Appendix C.3.3 for details). For datasets lacking group labels or explicit information about spurious features, an Open-Vocabulary Semantic Segmentation model can be applied, as in Appendix A.

**Dimensionality Reduction:** Since we are dealing with high dimensional image data, our next module compresses them into lower-dimensional discrete vectors. We propose to use an autoencoder, a deep neural network consisting of an encoder and a decoder, as shown in Fig. 6 to jointly do dimensionality reduction and clustering. We incorporate a bottleneck structure from (Sadeghi & Armanfard, 2023) in the encoder and decoder to obtain more informative lower-dimensional representation of the input image (see Fig. 17 in Appendix C). Along the lines of Guo et al. (2017), we obtain the clusters of the low-dimensional data $q$ by optimizing a joint loss function defined as $L = L_r + \gamma L_c$ where $L_r$ is the representation loss, $L_c$ is the clustering loss, and $\gamma$ is a non-negative

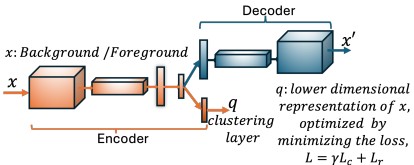

Figure 6: Dimensionality reduction module: Autoencoder with clustering to have discrete lower-dimensional embedding.

constant. The representation loss is the mean square error between the input of the encoder $x$ and output of the decoder $x'$ defined as $L_r = \|x - x'\|_2^2$. The cluster centers $\{\mu_j\}_1^K$ (trainable weights of clustering layer) and embedded point $z_i$ (output of the encoder) are used to calculate the soft label $q_{ij} = \frac{(1+\|z_i - \mu_j\|^2)^{-1}}{\sum_j (1+\|z_i - \mu_j\|^2)^{-1}}$ where $q_{ij}$ is the $j$th entry of the soft label $q_i$, denoting the probability of $z_i$ belonging to cluster $\mu_j$. The clustering loss $L_c$ is the KL divergence between the soft assignments ($q_i$) and an auxiliary distribution ($p_i$). First, the autoencoder is pre-trained using only $L_r$ to initialize the auxiliary distribution and the cluster centers are initialized by performing k-means on the embeddings of all images. After pretaining, the cluster centers and autoencoder weights are updated with the joint loss $L$ iteratively while the auxiliary distribution is only updated after $T$ iterations.

**Estimation:** The final step includes the estimation of joint distribution and the PID values, also leading to the proposed measure $M_{sp}$. The joint distribution is obtained by computing normalized 3D histogram of the discrete clusters of foreground, background, and binary target variable. Then, the PID values are estimated from the joint distribution using the DIT package (James et al., 2018) which is a python package for discrete information theory. We use $I_{BROJA}$ (BROJA Information) developed in (Bertschinger et al., 2014) to compute PID which solves the convex optimization problem in Definition 1 and results in four non-negative terms, namely, $\text{Uni}(Y{:}B|F)$, $\text{Uni}(Y{:}F|B)$, $\text{Red}(Y{:}F, B)$, and $\text{Syn}(Y{:}F, B)$. We use them to calculate the measure $M_{sp} = \text{Uni}(Y{:}B|F) - \text{Uni}(Y{:}F|B) - \text{Syn}(Y{:}F, B)$.

## 4 EXPERIMENTS

To support our theoretical findings, we provide experimental results for different datasets with different variants to capture different types of sampling biases, i.e., unbalanced, class-balanced, group-balanced, and mixed background (addition and concatenation). Here, we illustrate how information is distributed in the core and spurious features and how we can relate the worst-group accuracy (W.G. Acc.) with the proposed measure of the spuriousness $M_{sp}$ of a dataset. We conduct experiments on four datasets: Waterbird (Wah et al., 2011), CelebA (Lee et al., 2020), Dominoes (Shah et al., 2020), and Spawrious (Lynch et al., 2023). We begin with using our explainability framework, namely Spuriousness Disentangler, on each dataset (with dataset-specific variations) to compute the PID values and $M_{sp}$. We fine-tune the pre-trained ResNet-50 (He et al., 2016) model and calculate the worst-group accuracy over all groups. More details of the experiments are in Appendix C. Also see Appendix A (automatic segmentation of features), and Appendix B (Tabular datasets).

**1. Waterbird:** The Waterbird dataset (Wah et al., 2011) is a popular spurious correlation benchmark. The task is to classify the type of the bird (waterbird = 1, landbird = 0). However, there exists spurious correlation between the backgrounds (water = 1, land = 0) and the labels (bird type). The two types of backgrounds and foregrounds result in total four groups (details in Appendix C.3.1).

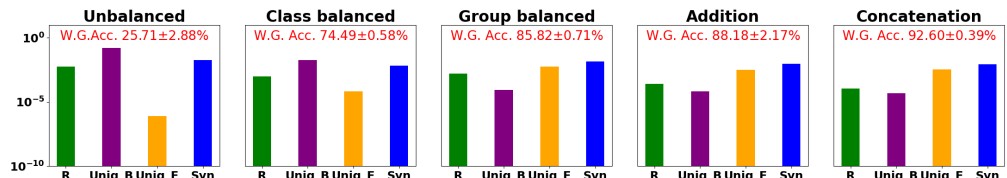

Figure 7: This bar-plot shows the redundant information (R), unique information in background (Uniq-B) and foreground (Uniq-F), and Synergistic information (Syn) for the Waterbird dataset for unbalanced, class balanced, group balanced, addition and concatenation setups. Observe that the Uniq-B decreases and Uniq-F increases for group balanced, addition, and concatenation dataset compared to that of unbalanced dataset. Note that the y-axis is in log scale.

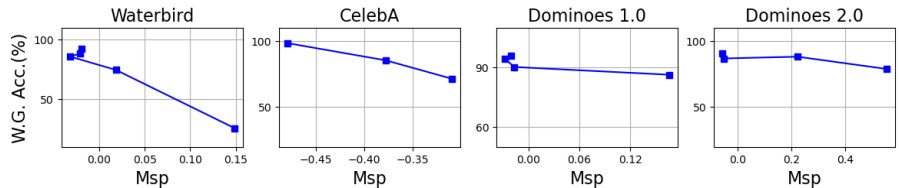

Figure 8: Trend between worst-group accuracy and measure of spuriousness $M_{sp}$ across datasets.

**Observations:** Fig. 7 shows our findings regarding PID values and the worst-group accuracy for the five variants of the Waterbird dataset. *Firstly,* we can observe that the unique information in background (Uniq-B) is significantly higher than the other PID values for unbalanced and class balanced cases. We also find an increase in unique information in foreground (Uniq-F) for the group balanced and background mixed versions. *Secondly*, the worst-group accuracy increases when any

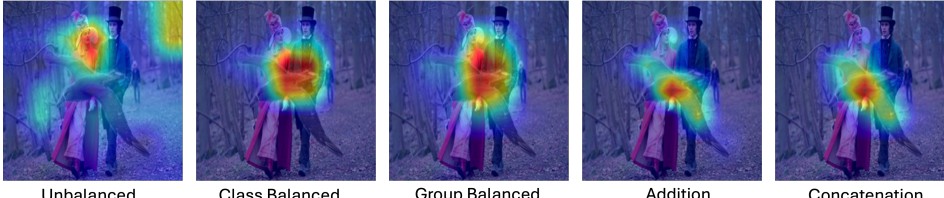

| Unbalanced | Class Balanced | Group Balanced | Addition | Concatenation |

Figure 9: Examples of Grad-CAM images of Waterbird dataset: Observe that for the unbalanced dataset (1st from left), the model adds more emphasis (red regions) to the background while in the class balanced, group balanced, addition and concatenation versions (2nd, 3rd, 4th and 5th from left), the foreground gets more emphasis.

technique is applied to reduce the sampling bias, namely, group balancing or background mixing. Fig. 8 depicts a negative trend between worst-group accuracy and measure of spuriousness $M_{sp}$. *Individual PID values do not give a complete understanding of the spuriousness, i.e., dataset's undesirable steering towards spurious features over core features. However the negative correlation between our proposed dataset measure $M_{sp}$ and the model generalization metric worst-group accuracy indicates that $M_{sp}$ is a good measure of dataset quality. Finally,* Fig. 9 shows through the Grad-CAM (Selvaraju et al., 2017) images that when the dataset is balanced or mixed background, the model emphasizes more on the core features (the red regions) while in the unbalanced dataset, the background is more emphasized which results in poor worst-group accuracy.

**2. CelebA:** CelebA is an another popular dataset for spurious correlation benchmarking which consists of images of male-female celebrities. We use a subset of this dataset namely CelebAMask-HQ (Lee et al., 2020) to utilize the segmentation mask of the hair while calculating the PID values. The objective is to identify blonde ($= 1$) and non-blonde ($= 0$) hair. However, there exists a spurious correlation between the gender (men ($= 1$), women ($= 0$)) and the label which makes the model focus on the face rather than the hair to find out the hair color (Moayeri et al., 2023). For this, we consider hair as the foreground and anything but the hair as background. We do not perform background mixing for this dataset since it is not practical to add or concatenate two faces randomly. More details are in Appendix C.3.2.

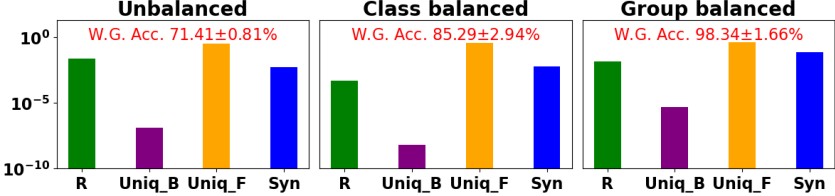

Figure 10: The distribution of the redundant information (R), unique information in background (Uniq-B) and foreground(Uniq-F) and Synergistic information (Syn) for the unbalanced, class balanced, and group balanced CelebA dataset. Observe that the Uniq-F and Synergy increase for class balanced and group balanced dataset compared to that of unbalanced dataset. Note that the y-axis is in log scale.

**Observations:** Fig. 10 shows the PID values for unbalanced, class balanced, and group balanced CelebA dataset. *Firstly,* the unique information in the foreground is the most prominent one among all other PID values. Observe that, the Uniq-F increases while the dataset is class balanced or group balanced along with the increasing worst-group accuracy. There is a negative trend between worst-group accuracy and the measure of spuriousness $M_{sp}$ (see Fig. 8). *Secondly,* the Grad-CAM images (see Fig. 21 in Appendix. C.3.2) show that the model focuses on the hair for the balanced dataset, but for the unbalanced dataset, it emphasizes more on the face.

**3. Dominoes:** Dominoes is a synthetic dataset created by combining handwritten digits (zero and one) from MNIST (Deng, 2012) and images of cars and trucks from CIFAR10 (Krizhevsky et al., 2009) (digit 0 or 1 at the top, car ($= 0$) or truck ($= 1$) at the bottom of an image). We make two version of this synthetic dataset namely Dominoes $1.0$ and Dominoes $2.0$ inducing different degrees

of sampling biases. The task is to classify whether the image contains a car or a truck; hence the car or truck corresponds to the core features (foreground). On the other hand, the digits are considered as the spurious features (background) (details in Appendix C.3.3).

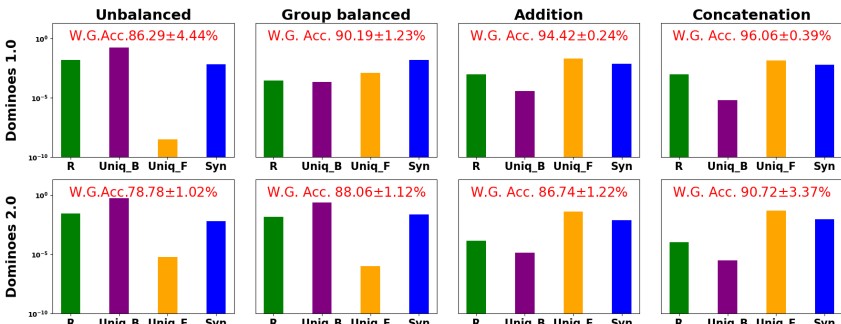

Figure 11: The distribution of the redundant information (R), unique information in background (Uniq-B) and foreground(Uniq-F) ,and Synergistic information (Syn) for the unbalanced, group balanced, addition and concatenation Dominoes dataset. Observe that the uniq-B decreases group balanced and background mixed datasets and the uniq-F increases for background mixed datasets compared to that of unbalanced dataset. Note that the y-axis is in log scale.

**Observations:** Fig. 11 shows the PID values for all four variations of Dominoes dataset. *Firstly,* the unique information in the background is really high for the unbalanced dataset. When the dataset is balanced or background mixed, this value decreases significantly. For the addition and concatenation cases, we observe that unique information in the foreground becomes significant. The worst-group accuracy improves when the training dataset is balanced or background mixed. *Secondly,* Fig. 8 shows a negative relationship between the worst-group accuracy and the spuriousness $M_{sp}$ for this dataset as well. *Finally,* in Fig. 22 of Appendix. C.3.3, we observe that the model focuses on the core features when there is reduced spuriousness, e.g., when the training dataset is balanced or background mixed.

**4. Spawrious:** Spawrious (Lynch et al., 2023) is a synthetic image dataset created by employing a text-to-image model. We use a subset of this dataset where we classify dog breeds - dachshund ($= 0$) and labrador ($= 1$). We select the subset in a way that most of the dachshunds are in beach ($= 0$) background and rest of them are in desert ($= 1$) background (see Appendix. C.3.4 for more details). We use a segmentation model with FPN (Lin et al., 2017) encoder and ResNet-$34$ (He et al., 2016) decoder pre-trained with Oxford-IIIT Pet Dataset to create the segmentation mask of the dogs of our dataset. Using this mask we separate the foreground "dog" from the background. After having the backgrounds and foregrounds, we use principal components analysis (PCA) (Maćkiewicz & Ratajczak, 1993) followed by k-means clustering to have discrete lower dimensional representation. We do not use our autoencoder module since for this dataset because a simpler dimensionality reduction also seems to have a low reconstruction loss.

**Observations** Fig. 12 shows that the redundancy and unique information in the background decrease and unique information in the foreground and synergy increase when the dataset is group balanced. We also observe that there is still a negative trend between the measure of spuriousness and the worst-group accuracy showing the effectiveness of the measure.

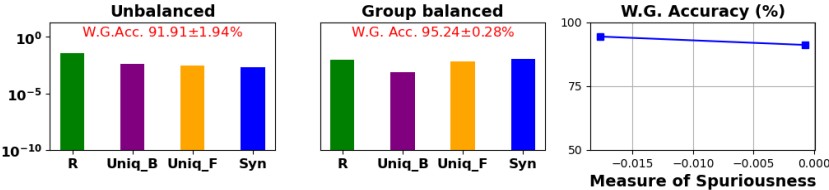

Figure 12: The first two plot shows the change in redundancy, unique information, and the synergistic information. The last plot shows a negative relationship between the worst-group accuracy and the measure of spuriousness $M_{sp}$. Note that the y-axis of first two subplots is in log scale.

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

# A ADDITIONAL EXPERIMENT 1: AUTOMATIC SEGMENTATION OF FEATURES

Segmentation, a component of our Spurious Disentangler, plays a pivotal role in identifying core features from spurious ones. Identifying spurious features (pixels) without any additional information is challenging in image datasets, particularly if they lack group labels. However, in supervised classification tasks, the availability of target labels corresponding directly to the goal of the classification task (and hence some partial knowledge of what the core features should be if not the exact pixels) often offers a practical workaround. Specifically, one can leverage automatic segmentation to at least perform object detection and choose the most relevant objects as the "core". Then, the regions of an image not associated with the "core" objects can often be considered a subset of spurious features.

Advances in Open-Vocabulary Semantic Segmentation (OVSS) have significantly reduced the dependence on task-specific training by enabling generalization to unseen categories without requiring labeled data. To leverage these advancements, we employ CLIPSeg (Lüddecke & Ecker, 2022), a state-of-the-art OVSS model, to generate masks for various objects in a zero-shot manner using partial knowledge of the classification task in mind. For instance, in the Waterbird dataset, we specify the prompt "bird" to obtain a mask for the bird object. This approach utilizes publicly available fine-grained weights, enabling efficient and accurate segmentation without additional labeled data.

The generated mask is applied to the original image to extract the foreground, while the background is obtained by multiplying the original image with $1 - mask$, as illustrated in Fig. 13. Fig. 14 reveals a negative correlation between the worst-group accuracy and increasing values of $M_{sp}$, calculated using the obtained background and foreground.

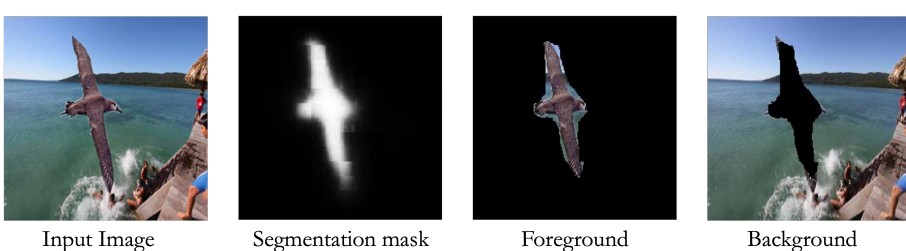

| Input Image | Segmentation mask | Foreground | Background |

Figure 13: The segmentation mask is obtained by zero-shot image segmentation using CLIPSeg (Lüddecke & Ecker, 2022). We get the foreground by multiplying the input image with the $mask$ and background by multiplying $(1 - mask)$.

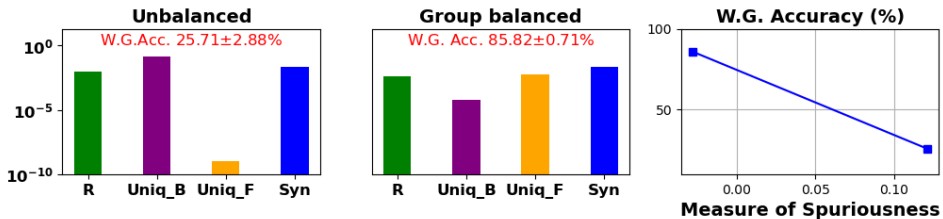

Figure 14: Waterbird Dataset: The first two plots show the change in redundancy, unique information, and the synergistic information. The last plot shows a negative relationship between the worst-group accuracy and the measure of spuriousness $M_{sp}$. Note that the y-axis of the first two plots is in log scale.

Thus, our proposed technique of dataset evaluation can be applied in conjunction with such automatic segmentation methods to any image dataset where the group information is not available, enabling us to first identify an approximation of the core features using partial knowledge of the target objects for the classification task, and then explain the nature of spurious patterns.

## B    ADDITIONAL EXPERIMENT 2: TABULAR DATASET

The applicability of our proposed framework goes beyond images, and can also be applied for explainability on tabular datasets. For instance, one might want to understand and interpret the dependencies of any specific feature with respect to another set of features in the dataset, prior to training. We perform an experiment on the Adult (Becker & Kohavi, 1996) dataset. The task is to predict whether annual income of an individual exceeds \$50k per year or not ($> 50k = 1, <= 50k = 0$). Here we consider "gender" as a spurious feature vector (male $= 1$, female $= 0$) and "age", "education-num", "hours-per-week" jointly as core feature matrix. Since the core feature matrix is high dimensional, we use k-means clustering to reduce the dimension and discretize the features. Then we use estimation module to calculate PID values with core features, spurious features, and target label. Fig. 15 shows the values for redundancy, unique information, and synergy. Observe that the unique information in the background and the redundant information approach zero, indicating that the correlation between gender and the target label has been effectively mitigated. We also observe a negative relationship between our proposed measure and worst-group accuracy which implies that $M_{sp}$ corresponds to the quality of the dataset. Furthermore, we observe a negative correlation between the proposed spurious measure $M_{sp}$ and the worst-group accuracy, highlighting that $M_{sp}$ serves as an indicator of dataset quality, i.e., spuriousness prior to training.

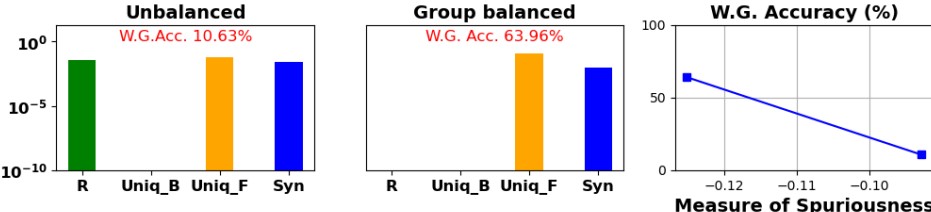

Figure 15: Adult Dataset: The first two plots show the PID values. The last plot shows a negative relationship between the worst-group accuracy and the measure of spuriousness $M_{sp}$. Note that the y-axis of first two subplots is in log scale.

We train XGBoost (Chen & Guestrin, 2016) model for prediction task and calculated the worst-group accuracy which corresponds to the accuracy of the minority group (see Table 1, minority group 10 corresponds to female individuals with >50k income.).

Table 1: Summary of Adult dataset

| Adult | Group 00 | Group 01 | Group 10 | Group 11 |
|-------|----------|----------|----------|----------|
| Train | 10116 | 15930 | 1214 | 6929 |
| Test | 4307 | 6802 | 555 | 2989 |
| Total | 14423 | 22732 | 1769 | 9918 |

## C    APPENDIX TO EXPERIMENTS

This section includes additional results and figures for a more comprehensive understanding.

### C.1    ADDITIONAL RESULTS

Our explainability framework is pre-emptive or anticipative of spuriousness using just the dataset before training the model. The goal of our experiments is to show broad agreement between our anticipations from the dataset before training any model and the post-training behavior of actual models (when trained regularly to optimize performance without doing anything else specifically targeted towards avoiding spurious features). Apart from Worst-Group Accuracy, we also observe the Grad-CAM visualizations to check if the model demonstrates a stronger emphasis on the relevant core features or not (see Fig. 9, 21, 22). To further justify this, we calculate intersection-over-union (IoU) metric (Rezatofighi et al., 2019) over the entire test Waterbird dataset. Table 2 shows that when

the dataset is modified from unbalanced to the other variants, the IoU score increases. The IoU score is calculated using the ground-truth segmentation masks of birds and the masks obtained from the Grad-CAM explanation.

Table 2: IoU between the ground truth masks and Grad-CAM masks for Waterbird dataset

| Test Dataset | Unbalanced | Class Balanced | Groupd Balanced | Addition | Concatenation |
|---|---|---|---|---|---|
| Minority Group | 0.22 | 0.29 | 0.24 | 0.28 | 0.32 |
| All Groups | 0.19 | 0.23 | 0.22 | 0.29 | 0.30 |

Table 3 shows a comparison between our proposed measure of spuriousness $M_{sp}$ and other possible measures.

Table 3: Comparison of our proposed measure of spuriousness $M_{sp}$ with other possible measures.

| Dataset | Measures | Unbalanced | Class Balanced | Group Balanced | Addition | Concatenation |
|---|---|---|---|---|---|---|
| | $I(Y;B)$ | 0.1726 | 0.0315 | 0.0028 | 0.0005 | 0.0002 |
| Waterbird | $I(Y;B) - I(Y;F)$ | 0.1669 | 0.0298 | -0.0089 | -0.0052 | -0.0054 |
| | Proposed $M_{sp}$ | 0.1486 | 0.0185 | -0.0322 | -0.0208 | -0.0195 |
| | $I(Y;B)$ | 0.1882 | - | 0.0005 | 0.0010 | 0.0010 |
| Dominoes 1.0 | $I(Y;B) - I(Y;F)$ | 0.1728 | - | -0.0010 | -0.0203 | -0.0144 |
| | Proposed $M_{sp}$ | 0.1660 | - | -0.0165 | -0.0279 | -0.0207 |
| | $I(Y;B)$ | 0.5913 | - | 0.2610 | 0.0002 | 0.0001 |
| Dominoes 2.0 | $I(Y;B) - I(Y;F)$ | 0.5619 | - | 0.2462 | -0.0426 | -0.0477 |
| | Proposed $M_{sp}$ | 0.5557 | - | 0.2237 | -0.0501 | -0.0574 |
| | $I(Y;B)$ | 0.0238 | 0.0005 | 0.0151 | - | - |
| CelebA | $I(Y;B) - I(Y;F)$ | -0.3038 | -0.3713 | -0.4051 | - | - |
| | Proposed $M_{sp}$ | -0.3091 | -0.3775 | -0.4797 | - | - |
| | $I(Y;B)$ | 0.0437 | - | 0.0096 | - | - |
| Spawrious | $I(Y;B) - I(Y;F)$ | 0.0012 | - | -0.0056 | - | - |
| | Proposed $M_{sp}$ | -0.0007 | - | -0.0176 | - | - |

## C.2 ADDITIONAL DETAILS ON CLUSTERING

At the dimensionality reduction step, we need to choose the number of clusters. We calculate the PID values for cluster number 5, 10, and 20. In Table 4, we observe that the relevant information can be preserved while reducing the dimensionality. We select 10 clusters to have a balance between retaining sufficient information and ensuring faster computational time.

Table 4: PIDs for Waterbird dataset with different number of clusters.

| Unbalanced | Red($Y$:$F$, $B$) | Uni($Y$:$B|F$) | Uni($Y$:$F|B$) | Syn($Y$:$F$, $B$) |
|---|---|---|---|---|
| # Cluster 5 | 0.0065 | 0.1220 | 0.0000 | 0.0085 |
| # Cluster 10 | 0.0057 | 0.1669 | 0.0000 | 0.0184 |
| # Cluster 20 | 0.0025 | 0.1736 | 0.0000 | 0.0163 |
| Standard Deviation | 0.0017 | 0.0229 | 0.0000 | 0.0043 |
| Class Balanced | Red($Y$:$F$, $B$) | Uni($Y$:$B|F$) | Uni($Y$:$F|B$) | Syn($Y$:$F$, $B$) |
| # Cluster 5 | 0.0008 | 0.0221 | 0.0000 | 0.0012 |
| # Cluster 10 | 0.0016 | 0.0300 | 0.0001 | 0.0114 |
| # Cluster 20 | 0.0008 | 0.0128 | 0.0000 | 0.0097 |
| Standard Deviation | 0.0004 | 0.0070 | 0.0000 | 0.0045 |

## C.3 ADDITIONAL DETAILS ON DATASETS

### C.3.1 WATERBIRD

A summary of the Waterbird dataset is given in Table 6. At first we use Spurious Disentangler for calculating PID values. The segmentation masks of the birds are given with the dataset. We multiply the given mask of each image with the corresponding whole image and get the foreground, i.e., the bird with black background and the backgrounds also come with the dataset (see Fig. 16 for the

Table 5: Worst-group accuracy(%) for different datasets with standard deviations.

| Dataset | Unbalanced | Class Balanced | Group Balanced | Addition | Concatenation |
|---|---|---|---|---|---|
| Waterbird | 25.71±2.88 | 74.49±0.58 | 85.82±0.71 | 88.18±2.17 | 92.60±0.39 |
| Dominoes 1.0 | 86.29±4.44 | - | 90.19±1.23 | 94.42±0.24 | 96.06±0.39 |
| Dominoes 2.0 | 78.78±1.02 | - | 88.06±1.12 | 86.74±1.22 | 90.72±3.37 |
| CelebA | 71.41±0.81 | 85.29±2.94 | 98.34±1.66 | - | - |
| Spawrious | 91.91±1.94 | - | 95.24±0.28 | - | - |

Table 6: Summary of the Waterbird dataset

| **Waterbird** | Group 00 | Group 01 | Group 10 | Group 11 |
|---|---|---|---|---|
| Train | 3498 | 184 | 56 | 1057 |
| Validation | 467 | 466 | 133 | 133 |
| Test | 2255 | 2255 | 642 | 642 |
| Total | 6220 | 2905 | 831 | 1832 |

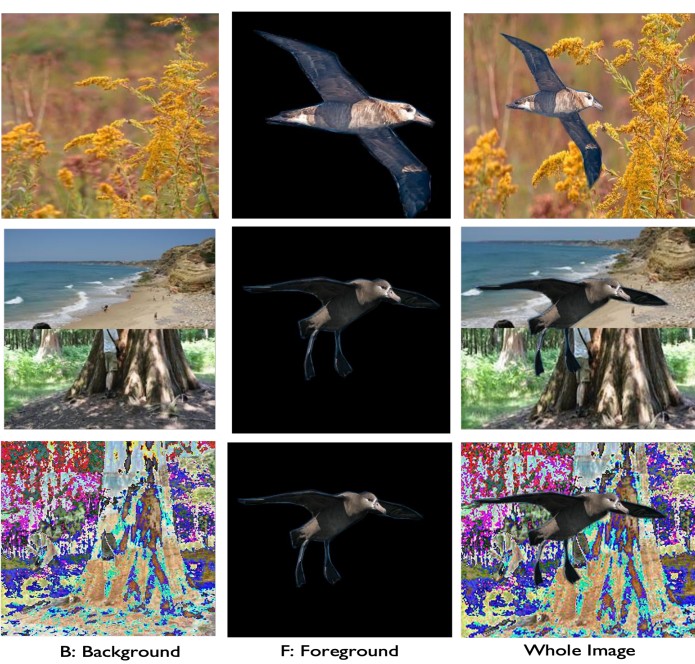

B: Background    F: Foreground    Whole Image

Figure 16: Samples of Waterbird dataset (original, concatenation, and addition).

examples of the dataset). For dimensionality reduction, we use autoencoder jointly with clustering as shown in Fig. 17. To obtain the clusters, the model is pre-trained with only mean square error loss function (MSEloss). Then, the model is again trained with weighted loss function which is a weighted sum of MSEloss and KL divergence loss with $\gamma = 0.1$ where the hyperparameter is chosen from standard implementations (Guo et al., 2017). The weights of the clustering layer are initialized with the cluster centers obtained by k-means clustering after the pre-training step. The training process is terminated if the change of label assignments between two consecutive updates for target distribution is less than $0.01$. The hyperparameters are as follows: a batch size of $64$, a learning rate of $0.001$, a CosineAnnealingLR scheduler, an Adam optimizer with a weight decay of $0.0001$, $150$ pretraining epochs, followed by $50$ epochs of additional training. Next, the clusters of the foreground, background, and the binary labels are used to estimate the joint distribution using 3D histograms followed by the PID estimation with DIT James et al. (2018) package which uses BROJA Information. See Table 7 for the details of PID values.

To calculate the worst-group accuracy we do fine-tuning of the pre-trained ResNet-50 He et al. (2016) model. The worst-group accuracy is defined as the accuracy of the minority group having the lowest

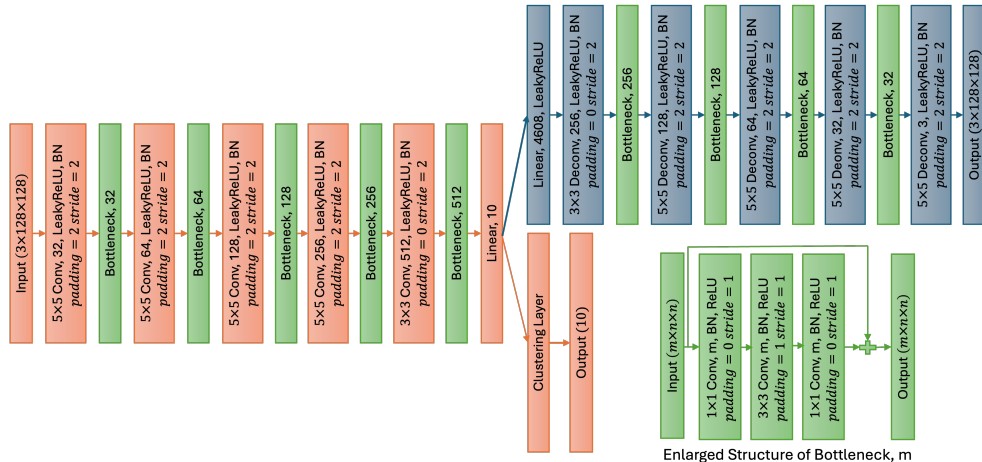

Figure 17: Architecture of the proposed autoencoder for the Waterbird and CelebA dataset. Here, BN stands for Batch Normalization.

Table 7: PID values for Waterbird dataset

| Waterbird | $\text{Red}(Y{:}F, B)$ | $\text{Uni}(Y{:}B\|F)$ | $\text{Uni}(Y{:}F\|B)$ | $\text{Syn}(Y{:}F, B)$ | $M_{sp}$ |
|---|---|---|---|---|---|
| Unbalanced | 0.0057 | 0.1669 | 0.0000 | 0.0184 | 0.1486 |
| Class Balanced | 0.0016 | 0.0300 | 0.0001 | 0.0114 | 0.0185 |
| Group Balanced | 0.0026 | 0.0001 | 0.0091 | 0.0233 | -0.0322 |
| Addition | 0.0004 | 0.0001 | 0.0053 | 0.0156 | -0.0208 |
| Concatenation | 0.0002 | 0.0001 | 0.0055 | 0.0140 | -0.0195 |

number of training sample. The worst-group accuracy is defined as the accuracy of the minority group with the fewest training samples. The hyperparameters used are as follows: batch size of 64, learning rate of 0.0001, CosineAnnealingLR scheduler, stochastic gradient descent (SGD) optimizer with a weight decay of 0.0001, binary cross-entropy as the loss function, and 100 epochs. For balanced datasets, we use a weighted random sampler, where the weights are selected based on the proportion of the groups or classes. See Table 5 for the worst-group accuracies of different variants of Waterbird dataset.

### C.3.2 CELEBA

Table 8: Summary of the CelebA dataset

| **CelebA** | Group 00 | Group 01 | Group 10 | Group 11 |
|---|---|---|---|---|
| Train | 11111 | 8305 | 4003 | 188 |
| Test | 1391 | 997 | 525 | 18 |
| Total | 12502 | 9302 | 4528 | 206 |

The summary of the CelebA (Lee et al., 2020) dataset is given in Table 8. The steps and hyperparameters for calculating PIDs are same as Waterbird dataset. However, we get the background, by multiplying (1-mask) with the whole image. See Fig. 18 for the examples.

Table 9: PID values for CelebA dataset

| CelebA | $\text{Red}(Y{:}F, B)$ | $\text{Uni}(Y{:}B\|F)$ | $\text{Uni}(Y{:}F\|B)$ | $\text{Syn}(Y{:}F, B)$ | $M_{sp}$ |
|---|---|---|---|---|---|
| Unbalanced | 0.0238 | 0.0000 | 0.3038 | 0.0053 | -0.3091 |
| Class Balanced | 0.0005 | 0.0000 | 0.3713 | 0.0063 | -0.3775 |
| Group Balanced | 0.0151 | 0.0000 | 0.4051 | 0.0746 | -0.4797 |

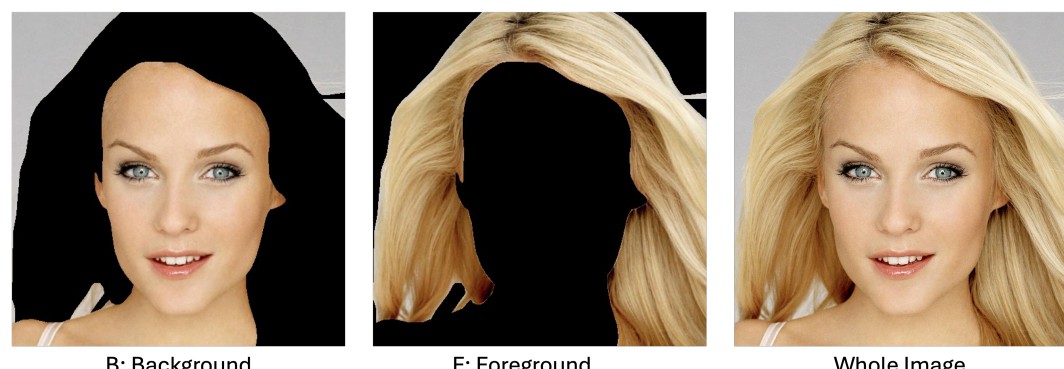

B: Background                F: Foreground                Whole Image

Figure 18: Samples of CelebA dataset.

The details of PID values and worst-group accuracies for several variations of this dataset are shown in Table 9 and Table 5 respectively.

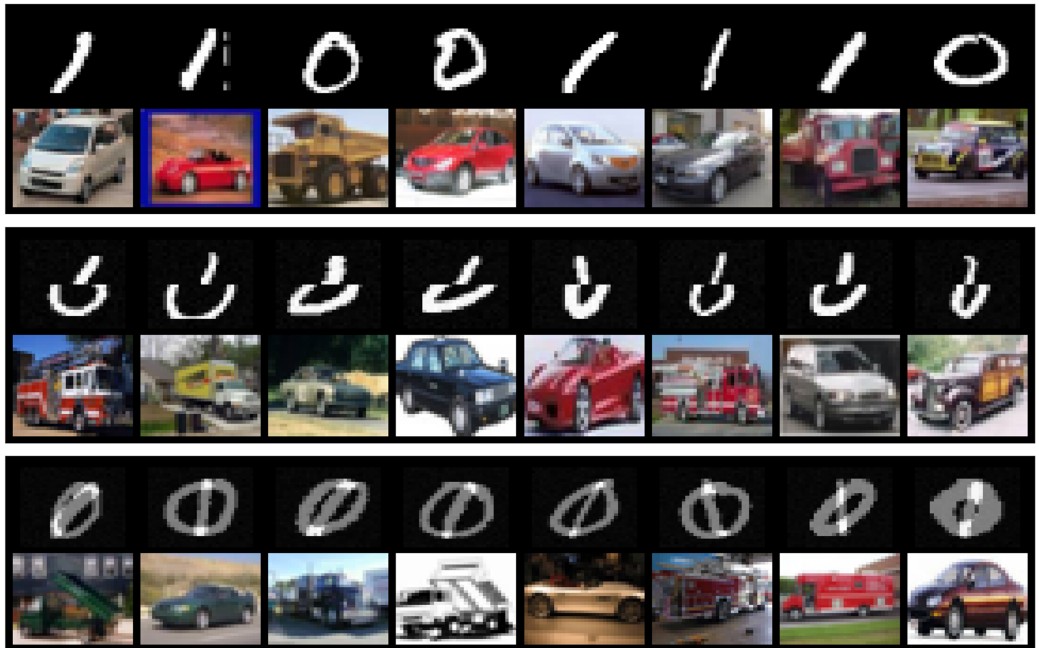

Figure 19: Samples of Dominoes dataset (original, concatenation, and addition).

### C.3.3  DOMINOES

The summary of Dominoes 1.0 and Dominoes 2.0 are given in Table 10 and Table 11 respectively. Fig. 19 shows the examples of original, addition, and concatenation variants of the dataset.

Table 10: Summary of the Dominoes 1.0 dataset

| Dominoes 1.0 | Group 00 | Group 01 | Group 10 | Group 11 |
|---|---|---|---|---|
| Train | 3750 | 1250 | 1250 | 3750 |
| Test | 473 | 507 | 507 | 473 |
| Total | 4223 | 1772 | 1757 | 4208 |

Table 11: Summary of the Dominoes 2.0 Dataset

| **Dominoes 2.0** | Group 00 | Group 01 | Group 10 | Group 11 |
|---|---|---|---|---|
| Train | 3000 | 500 | 1250 | 3000 |
| Test | 245 | 490 | 245 | 490 |
| Total | 3245 | 990 | 1495 | 3490 |

Table 12: PID values for Dominoes dataset

| Dominoes 1.0 | $\text{Red}(Y{:}F,B)$ | $\text{Uni}(Y{:}B\vert F)$ | $\text{Uni}(Y{:}F\vert B)$ | $\text{Syn}(Y{:}F,B)$ | $M_{sp}$ |
|---|---|---|---|---|---|
| Unbalanced | 0.0154 | 0.1728 | 0.0000 | 0.0068 | 0.1660 |
| Group Balanced | 0.0003 | 0.0002 | 0.0013 | 0.0155 | -0.0165 |
| Addition | 0.0009 | 0.0000 | 0.0203 | 0.0076 | -0.0279 |
| Concatenation | 0.0009 | 0.0000 | 0.0144 | 0.0063 | -0.0207 |
| Dominoes 2.0 | $\text{Red}(Y{:}F,B)$ | $\text{Uni}(Y{:}B\vert F)$ | $\text{Uni}(Y{:}F\vert B)$ | $\text{Syn}(Y{:}F,B)$ | $M_{sp}$ |
| Unbalanced | 0.0294 | 0.5619 | 0.0000 | 0.0061 | 0.5557 |
| Group Balanced | 0.0148 | 0.2462 | 0.0000 | 0.0225 | 0.2237 |
| Addition | 0.0001 | 0.0000 | 0.0426 | 0.0075 | -0.0501 |
| Concatenation | 0.0001 | 0.0000 | 0.0477 | 0.0096 | -0.0574 |

For PID calculation, The hyperparameters are as follows: a batch size of $8$, a learning rate of $0.001$, a CosineAnnealingLR scheduler, an Adam optimizer with a weight decay of $0.0001$, 100 pretraining epochs, followed by 50 epochs of additional training. The architecture of the autoencoder is given in Table 13. See Table 12 for the details of PID values and $M_{sp}$. For Dominoes 1.0 dataset, since group 01 and group 10 have the same number of training and test samples, the worst-group accuracy is calculated by taking the average of the accuracies of these two groups. Table 5 shows the worst-group accuracies for unbalanced, group balanced, addition, and concatenation datasets.

Table 13: Architecture details of autoencoder for Dominoes dataset

| Sl. No. | Layer | Filter No. | Kernel Size | Stride | Padding | Output Padding | Output Shape | Param No. |
|---|---|---|---|---|---|---|---|---|
| 1 | Conv2d | 32 | 5 | 2 | 2 | - | (32,16,16) | 2432 |
| 2 | LeakyReLU | - | - | - | - | - | (32,16,16) | 0 |
| 3 | BatchNorm2d | - | - | - | - | - | (32,16,16) | 64 |
| 4 | Conv2d | 64 | 5 | 2 | 2 | - | (64,8,8) | 51264 |
| 5 | LeakyReLU | - | - | - | - | - | (64,8,8) | 0 |
| 6 | BatchNorm2d | - | - | - | - | - | (64,8,8) | 128 |
| 7 | Conv2d | 128 | 3 | 2 | 0 | - | (128,3,3) | 73856 |
| 8 | LeakyReLU | - | - | - | - | - | (128,3,3) | 0 |
| 9 | Flatten | - | - | - | - | - | 1152 | 0 |
| 10 | Linear (embedding) | - | - | - | - | - | 10 | 11530 |
| 11 | Clustering Layer | - | - | - | - | - | 10 | 100 |
| 12 | Linear(deembedding) | - | - | - | - | - | 1152 | 12672 |
| 13 | LeakyReLU | - | - | - | - | - | 1152 | 0 |
| 14 | ConvTranspose2d | 64 | 3 | 2 | 0 | 1 | (64, 8, 8) | 73,792 |
| 15 | LeakyReLU | - | - | - | - | - | (64, 8, 8) | 0 |
| 16 | BatchNorm2d | - | - | - | - | - | (64, 8, 8) | 128 |
| 17 | ConvTranspose2d | 32 | 5 | 2 | 2 | 1 | (32, 16, 16) | 51,232 |
| 18 | LeakyReLU | - | - | - | - | - | (32, 16, 16) | 0 |
| 19 | BatchNorm2d | - | - | - | - | - | (32, 16, 16) | 64 |
| 20 | ConvTranspose2d | 3 | 5 | 2 | 2 | 1 | (3, 32, 32) | 2403 |

### C.3.4 SPAWRIOUS

The summary of the subset of Spawrious dataset Lynch et al. (2023) that we use for our experiment is given in Table 14. The samples of this dataset are shown in Fig. 20.

Table 14: Summary of the subset of the Spawrious dataset

| **Spawrious** | Group 00 | Group 01 | Group 10 | Group 11 |
|---|---|---|---|---|
| Train | 3072 | 2275 | 175 | 1056 |
| Test | 96 | 893 | 2993 | 2112 |
| Total | 3168 | 3168 | 3168 | 3168 |

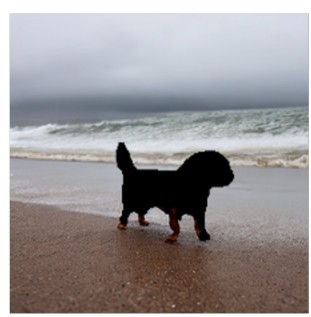 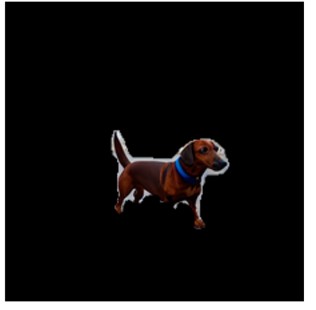 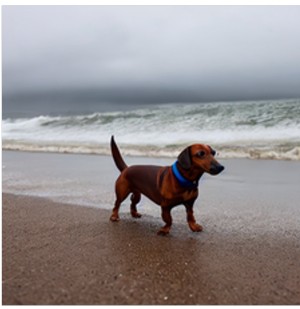

B: Background          F: Foreground          Whole Image

Figure 20: Samples of the subset of Spawrious dataset we use in this work.

Table 15: PID values for Spawrious dataset

| Spawrious | $\mathrm{Red}(Y{:}F,B)$ | $\mathrm{Uni}(Y{:}B|F)$ | $\mathrm{Uni}(Y{:}F|B)$ | $\mathrm{Syn}(Y{:}F,B)$ | $M_{sp}$ |
|---|---|---|---|---|---|
| Unbalanced | 0.039589 | 0.004067 | 0.002883 | 0.001921 | -0.00074 |
| Group Balanced | 0.00891 | 0.000699 | 0.006276 | 0.012068 | -0.01765 |

We use pre-trained segmentation model to generate the mask of the dog and separate the foreground and background using this mask. We use PCA followed by k-means clustering to have lower dimensional discrete representation of the foreground and background. Then we use our estimation module for the calculation of PID values and $M_{sp}$. Table 15 and Table 5 shows all PID values along with the measure and the worst-group accuracy respectively. All the experiments are executed on NVIDIA RTX A4500.

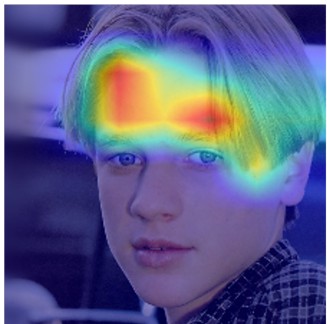 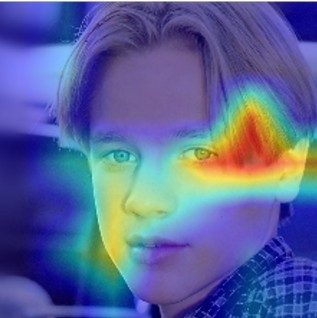 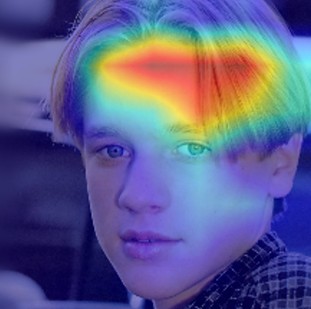

Figure 21: Examples of Grad-CAM images CelebA dataset: Observe that for the unbalanced dataset (1st from left), the model adds more emphasis (red regions) to the face (background) while in the class balanced and group balanced (2nd and 3rd), the hair (foreground) is more emphasized.

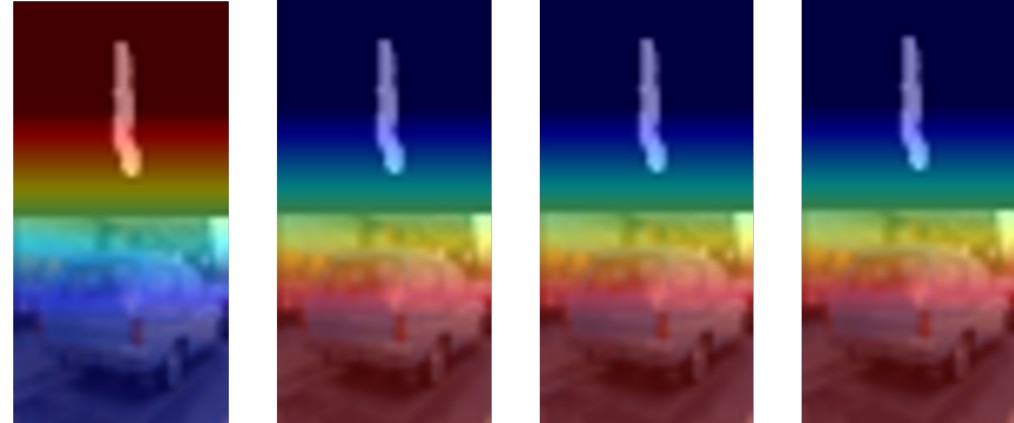

Figure 22: Examples of Grad-CAM images Dominoes dataset: Observe that for the unbalanced dataset (1st from left), the model adds more emphasis (red regions) to the digits (background) while in the group balanced, addition and concatenation versions (2nd, 3rd, and 4th from left), the car (foreground) is more emphasized.

## D    DISCUSSION AND FUTURE WORK

Formalizing and analyzing the information distribution among spurious and core features can provide a theoretical understanding of the biases or spuriousness of any dataset. Calculating the measure of spuriousness introduces an efficient way to assess dataset quality before performing the actual training or fine-tuning which can be computationally intensive, particularly in the era of foundational models. In this work, we theoretically justify the importance of each component of partial information decomposition (PID) for understanding the nature of dataset and the prediction. We also justify the validation of the proposed measure of spuriousness with examples and counterexamples along with experimental findings that proposed measure has a relationship with worst-group accuracy (and hence, dataset quality). We also introduce the use of Spurious Disentangler for handling high dimensional image data and estimating the PIDs (Broader Impacts in Appendix E).

**Limitations:** (i) Identifying spurious features and core features of a given dataset automatically is not always straightforward. Future work will look into alternate techniques, such as causal discovery (Zanga et al., 2022) (recently using LLMs (Liu et al., 2024)) as well as validation on NLP datasets. (ii) The estimation is highly data-dependent. A small change in the dataset can greatly affect the PID values. Future work will look into sensitivity and estimation error analysis. (iii) The efficiency and robustness of the Spurious Disentangler can also be improved. (iv) Additionally, there can be groups of spurious features rather than just one which can have nuanced interplay among them, which is another interesting direction.

## E    BROADER IMPACT

Quantifying spurious patterns has significant broader impacts across multiple domains. Quantification of dataset spuriousness might improve the trustworthiness of AI in several high-stakes and safety-critical applications such as healthcare which can directly impact people's lives. Spurious patterns often lead to biased predictions, particularly in sensitive domains such as hiring, lending, or criminal sentencing. Going beyond existing works, our research paves the way for improved understanding of the nature of spurious relationships, enabling interpretability which could also have significant implications in auditing and preventing discrimination.

# F  APPENDIX TO MAIN RESULTS

## F.1  RELEVANT MATHEMATICAL RESULTS

PID (Bertschinger et al., 2014; Banerjee et al., 2018) provides a mathematical framework that decomposes the total information content $I(Y; A, B)$ into four non-negative terms:

$$I(Y; A, B) = \text{Uni}(Y{:}B|A) + \text{Uni}(Y{:}A|B) + \text{Red}(Y{:}A, B) + \text{Syn}(Y{:}A, B). \quad (5)$$

In addition to this equation, the PID terms also satisfy the following relationships (Bertschinger et al., 2014; Banerjee et al., 2018):

$$I(Y; A) = \text{Uni}(Y{:}A|B) + \text{Red}(Y{:}A, B). \quad (6)$$

$$I(Y; A|B) = \text{Uni}(Y{:}A|B) + \text{Syn}(Y{:}A, B). \quad (7)$$

Now, defining any one of the PID terms is sufficient to obtain all four by using these relationships. In this work, we use a popular definition of unique information from (Bertschinger et al., 2014; Banerjee et al., 2018) as defined in Definition 1 in Section 2 which can be computed by solving a convex optimization problem (Bertschinger et al., 2014; Banerjee et al., 2018).

One of the most desirable property of this definition is that all four PID terms are non-negative.

**Lemma 4** (Nonnegativity of PID). *All four PID terms* $\text{Uni}(Y{:}B|A)$, $\text{Uni}(Y{:}A|B)$, $\text{Red}(Y{:}A, B)$, *and* $\text{Syn}(Y{:}A, B)$ *are nonnegative as per Definition 1.*

This result is proved in Bertschinger et al. (2014, Lemma 5).

**Lemma 5** (Monotonicity under local operations on B). *Let* $B = f(B')$ *where* $f(\cdot)$ *is a deterministic function. Then, we have:*

$$\text{Uni}(Y{:}B|A) \leq \text{Uni}(Y{:}B'|A).$$

This result is derived in Banerjee et al. (2018, Lemma 31).

**Lemma 6** (Monotonicity under adversarial side information). *For all* $(Y, B, A, W)$, *we have:*

$$\text{Uni}(Y{:}B|A, W) \leq \text{Uni}(Y{:}B|A).$$

This result is derived in Banerjee et al. (2018, Lemma 32).

**Lemma 7.** $\text{Uni}(Y{:}B|F) = 0$ *if and only if there exists a row-stochastic matrix* $T \in [0,1]^{|\mathcal{F}| \times |\mathcal{B}|}$ *such that:* $P_{YB}(Y = y, B = b) = \sum_{f \in \mathcal{F}} P_{YF}(Y = y, F = f) T(f, b)$ *for all* $y \in \mathcal{Y}$ *and* $b \in \mathcal{B}$.

*Proof.* This result is from Bertschinger et al. (2014). Here, we include a proof for completeness.

If $\text{Uni}(Y{:}B|F) = 0$, then we have: $\min_{Q \in \Delta_P} I_Q(Y; B|F) = 0$ where $\Delta_P = \{Q \in \Delta : Q_{YF}(Y = y, F = f) = P_{YF}(Y = y, F = f)$ and $Q_{YB}(Y = y, B = b) = P_{YB}(Y = y, B = b)\}$. Thus, there exists a distribution $Q \in \Delta_P$ such that $Y$ and $B$ are independent given $F$ under the joint distribution $Q$. Then, we have

$$P_{YB}(Y = y, B = b) = Q_{YB}(Y = y, B = b) \quad (8)$$

$$= \sum_{f \in \mathcal{F}} Q_{YFB}(Y = y, F = f, B = b) \quad (9)$$

$$= \sum_{f \in \mathcal{F}} Q_{B|YF}(B = b|Y = y, F = f) Q_{YF}(Y = y, F = f) \quad (10)$$

$$\stackrel{(a)}{=} \sum_{f \in \mathcal{F}} Q_{B|YF}(B = b|Y = y, F = f) P_{YF}(Y = y, F = f) \quad (11)$$

$$\stackrel{(b)}{=} \sum_{f \in \mathcal{F}} Q_{B|F}(B = b|F = f) P_{YF}(Y = y, F = f) \quad (12)$$

$$\stackrel{(c)}{=} \sum_{f \in \mathcal{F}} T(f, b) P_{YF}(Y = y, F = f). \quad (13)$$

Here, (a) holds because $P_{YF} = Q_{YF}$ for all $Q \in \Delta_P$, (b) holds because under joint distribution $Q$, variables $Y$ and $B$ are independent given $F$, and (c) simply chooses $T(f, b) = Q_{B|F}(B = b|F = f)$ which is a function of $(f, b)$ and will lead to a row-stochastic matrix $T$ since $\sum_{b \in \mathcal{B}} T(f, b) = \sum_{b \in \mathcal{B}} Q_{B|F}(B = b|F = f) = 1$.

Next, we prove the converse. Suppose, such a row-stochastic matrix $T$ exists such that:

$$P_{YB}(Y = y, B = b) = \sum_{f \in \mathcal{F}} T(f, b) P_{YF}(Y = y, F = f).$$

Now, we can define a joint distribution $Q^*$ such that:

$$Q^*(Y = y, F = f, B = b) = P_{YF}(Y = y, F = f) T(f, b). \tag{14}$$

We can show that $Q^*$ is a valid probability distribution since $T$ is row stochastic.

$$\sum_{y \in \mathcal{Y}} \sum_{b \in \mathcal{B}} \sum_{f \in \mathcal{F}} Q^*(Y = y, F = f, B = b) = \sum_{y \in \mathcal{Y}} \sum_{b \in \mathcal{B}} \sum_{f \in \mathcal{F}} P_{YF}(Y = y, F = f) T(f, b)$$

$$= \sum_{y \in \mathcal{Y}} \sum_{f \in \mathcal{F}} P_{YF}(Y = y, F = f) \left( \sum_{b \in \mathcal{B}} T(f, b) \right)$$

$$= \sum_{y \in \mathcal{Y}} \sum_{f \in \mathcal{F}} P_{YF}(Y = y, F = f) = 1. \tag{15}$$

Also, we can show that $Q^* \in \Delta_P$ since:

$$Q^*_{YB}(Y = y, B = b) = \sum_{f \in \mathcal{F}} P_{YF}(Y = y, F = f) T(f, b) = P_{YB}(Y = y, B = b), \tag{16}$$

which holds since such a row-stochastic matrix $T$ exists. Also, we have:

$$Q^*_{YF}(Y = y, F = f) = \sum_{b \in \mathcal{B}} P_{YF}(Y = y, F = f) T(f, b) = P_{YF}(Y = y, F = f), \tag{17}$$

which holds since $T$ is row-stochastic.

Then, $\mathrm{Uni}(Y{:}B|F) = \min_{Q \in \Delta_P} \mathrm{I}_Q(Y; B|F) \leq \mathrm{I}_{Q^*}(Y; B|F) = 0$.

$\square$

### F.2 PROOF OF THEOREM 1

For the first claim, notice that $\mathrm{Uni}(Y{:}B|F) = \mathrm{I}(Y; B) - \mathrm{Red}(Y{:}B, F)$ (from equation 6) and $\mathrm{Red}(Y{:}B, F) \geq 0$ (nonnegativity of PID, see Lemma 4). Thus,

$$\mathrm{Uni}(Y{:}B|F) \leq \mathrm{I}(Y; B).$$

For the second claim, we will use Lemma 7. $\mathrm{Uni}(Y{:}B|F) = 0$ if and only if there exists a row-stochastic matrix $T \in [0, 1]^{|\mathcal{F}| \times |\mathcal{B}|}$ such that: $P_{YB}(Y = y, B = b) = \sum_{f \in \mathcal{F}} P_{YF}(Y = y, F = f) T(f, b)$ for all $y \in \mathcal{Y}$ and $b \in \mathcal{B}$. The existence of such a row-stochastic matrix is equivalent to Blackwell Sufficiency as per Definition 2 from (Blackwell, 1953).

For the third claim, first observe that if $B' = B \cup W$, then $B$ can be written as a local operation on $B'$, i.e., $B = f(B')$. Thus, from Lemma 5, we have:

$$\mathrm{Uni}(Y{:}B|F) \leq \mathrm{Uni}(Y{:}B'|F). \tag{18}$$

Next, observe that since $F' = F \backslash W$, then from Lemma 6, we have:

$$\mathrm{Uni}(Y{:}B'|F) = \mathrm{Uni}(Y{:}B'|F', W) \leq \mathrm{Uni}(Y{:}B'|F'). \tag{19}$$

Combining equation 18 and equation 19, we have the claim

$$\mathrm{Uni}(Y{:}B|F) \leq \mathrm{Uni}(Y{:}B'|F').$$

### F.3 PROOF OF ADDITIONAL RESULTS

#### F.3.1 PROOF OF LEMMA 1

*Proof of Lemma 1.* Here, $B = Y + N$ and $F = Y + N$ where $Y$ and $N$ are independent. Any optimal predictor is a function of the inputs $F$ and $B$, i.e., $\hat{Y} = f(F, B)$. Since $F = B$, this function can always be rewritten as a function of $B$ alone or $F$ alone.

Next, we will show that only the redundant information $\text{Red}(Y{:}B, F)$ is positive and all other PID terms $\text{Uni}(Y{:}B|F)$, $\text{Uni}(Y{:}F|B)$, and $\text{Syn}(Y{:}F, B)$ are zero.

Here $I(Y; B|F) = I(Y; F|B) = 0$ since $B = F$.

$$I(Y; B|F) = H(B|F) - H(B|Y, F) = 0.$$

According to the Definition 1 and non-negativity of PID terms, $\text{Uni}(Y{:}B|F) = I(Y; B|F) - \text{Syn}(Y{:}F, B) \leq I(Y; B|F) = 0$.

Similarly, we have, $\text{Uni}(Y{:}F|B) \leq I(Y; F|B) = 0$.

Then, $\text{Syn}(Y{:}F, B) = I(Y; F|B) - \text{Uni}(Y{:}F|B)$ (from equation 7) is also 0.

Now, $\text{Red}(Y{:}B, F) = I(Y; B) - \text{Uni}(Y{:}B|F) = I(Y; B) = H(Y) - H(Y|B)$ which is positive as long as there is a significant dependence between $Y$ and $B$.

$\square$

#### F.3.2 PROOF OF LEMMA 2

We first include another lemma that will be useful in proving our main result.

**Lemma 8** (Noisy Feature). *Let $A = Y + N$ where $Y \sim Bern(1/2)$ is a random variable taking values $+1$ or $-1$ and the noise $N \sim \mathcal{N}(0, \sigma_N^2)$ is a Gaussian random variable independent of $Y$. Then, the mutual information*

$$I(Y; A) \leq \frac{1}{2} \log_2 \left( 1 + \frac{1}{\sigma_N^2} \right).$$

*Proof.*

$$I(Y; A) = H(A) - H(A|Y) = H(Y + N) - H(Y + N|Y) \tag{20}$$

$$= H(Y + N) - H(N|Y) \tag{21}$$

$$= H(Y + N) - H(N), \text{ since } N \perp\!\!\!\perp Y \tag{22}$$

$$\overset{(a)}{\leq} \frac{1}{2} \log_2 2\pi e \left( 1 + \sigma_N^2 \right) - \frac{1}{2} \log_2 2\pi e \left( \sigma_N^2 \right) \tag{23}$$

$$= \frac{1}{2} \log_2 \left( 1 + \frac{1}{\sigma_N^2} \right). \tag{24}$$

Here (a) holds because the entropy of $Y + N$ is bounded by $\frac{1}{2} \log_2 2\pi e \left( 1 + \sigma_N^2 \right)$ (proved in Cover & Thomas (2012, Theorem 8.6.5)). We also refer to Cover & Thomas (2012, Chapter 9) for a discussion on Gaussian channels. $\square$

If we keep the distribution of $Y$ fixed and vary the noise variance $\sigma_N^2$, then we will observe a decreasing trend of $I(Y; B)$ with increasing $\sigma_N^2$. Fig.23 shows the exact trend where $Y$ is a Bernoulli random variable.

*Proof of Lemma 2.* Here $B = N$ and $F = Y + N$ where $Y \sim Bern(1/2)$ takes values $+1$ or $-1$, and the noise $N \sim \mathcal{N}(0, \sigma_N^2)$ with $N \perp\!\!\!\perp Y$ and $\sigma_N^2 \gg 1$.

First observe that the predictor $\hat{Y} = f(B, F) = F - B = Y$. Thus, it is perfectly predictive of $Y$, and is an optimal predictor.

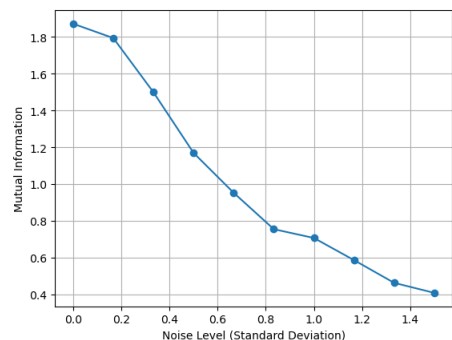

Figure 23: Mutual Information vs. Noise Level ($Y$ is Bernoulli)

Now, we will compute the values of the PID terms and show that $\mathrm{Syn}(Y{:}B, F) > 0$ and all the other three PID terms are negligible.

Since $B \perp\!\!\!\perp Y$, we have $\mathrm{I}(Y; B) = 0$.

Since $F = Y + N$, we use Lemma 8 to first show that: $\mathrm{I}(Y; F) \leq \frac{1}{2} \log_2 \left( 1 + \frac{1}{\sigma_N^2} \right)$. Now, as the variance $\sigma_N^2$ becomes high, we have $\mathrm{I}(Y; F) \approx 0$.

Since $N \perp\!\!\!\perp Y$, we have $\mathrm{I}(Y; B) = 0$. Now, from equation 6, we have

$$\mathrm{I}(Y; B) = \mathrm{Uni}(Y{:}B|F) + \mathrm{Red}(Y{:}B, F) = 0. \tag{25}$$

According to Lemma 4, $\mathrm{Uni}(Y{:}B|F)$ and $\mathrm{Red}(Y{:}B, F)$ are nonnegative. As their summation is 0, each term should be 0 as well, i.e., $\mathrm{Uni}(Y{:}B|F) = 0$ and $\mathrm{Red}(Y{:}B, F) = 0$.

Again, since $N$ has a high variance, we have (from Lemma 8):

$$\mathrm{I}(Y; F) \leq \frac{1}{2} \log_2 \left( 1 + \frac{1}{\sigma_N^2} \right) \approx 0. \tag{26}$$

This leads to $\mathrm{Uni}(Y{:}F|B) = \mathrm{I}(Y; F) - \mathrm{Red}(Y{:}B, F) \leq \frac{1}{2} \log_2 \left( 1 + \frac{1}{\sigma_N^2} \right) \approx 0$.

However, $\mathrm{I}(Y; F|B) = H(Y|B) - H(Y|B, F) = H(Y|N) - H(Y|Y + N, N) = H(Y)$ which is positive and significant. This holds because $H(Y|Y + N, N) = 0$ since $Y$ is completely determined by $Y + N$ and $N$ together.

Now,

$$\mathrm{Syn}(Y{:}B, F) = \mathrm{I}(Y; F|B) - \mathrm{Uni}(Y{:}F|B) \geq H(Y) - \frac{1}{2} \log_2 \left( 1 + \frac{1}{\sigma_N^2} \right) \approx H(Y). \tag{27}$$

$\square$

### F.3.3 PROOF OF LEMMA 3

*Proof of Lemma 3.* Here the input feature $X = (F, B)$. Observe that, we have the following conditional distributions: $X|_{Y=0} \sim \mathcal{N}([0 \ 0], \begin{bmatrix} \sigma_{N_F}^2 & 0 \\ 0 & \sigma_{N_B}^2 \end{bmatrix})$, and $X|_{Y=1} \sim \mathcal{N}([1 \ 1], \begin{bmatrix} \sigma_{N_F}^2 & 0 \\ 0 & \sigma_{N_B}^2 \end{bmatrix})$.

For simplicity, assume $P(Y = 0) = P(Y = 1)$. We let $\Sigma = \begin{bmatrix} \sigma_{N_F}^2 & 0 \\ 0 & \sigma_{N_B}^2 \end{bmatrix}$.

For the Bayes optimal classifier at the decision boundary, we have:

$$P(X|Y=0) = P(X|Y=1)$$
$$\Rightarrow \log(P(X|Y=0)) = \log(P(X|Y=1))$$
$$\Rightarrow -\frac{1}{2}X\Sigma^{-1}X^\top = -\frac{1}{2}(X - [1\ 1])\Sigma^{-1}(X - [1\ 1])^\top$$
$$\Rightarrow \frac{\|F\|_2^2}{\sigma_{N_F}^2} + \frac{\|B\|_2^2}{\sigma_{N_B}^2} = \frac{\|F-1\|_2^2}{\sigma_{N_F}^2} + \frac{\|B-1\|_2^2}{\sigma_{N_B}^2}$$
$$\Rightarrow \frac{F}{\sigma_{N_F}^2} + \frac{B}{\sigma_{N_B}^2} = \frac{1}{2\sigma_{N_F}^2} + \frac{1}{2\sigma_{N_B}^2}$$

This is the decision boundary for the Bayes optimal classifier. Thus, we can show that when $\sigma_{N_B}^2 \gg \sigma_{N_F}^2$, the boundary relies heavily on core feature $F$. Similarly, when $\sigma_{N_F}^2 \gg \sigma_{N_B}^2$, the boundary relies heavily on spurious feature $B$. Also refer to Fig. 3 (first two cases) for a pictorial illustration on how the optimal classifier behaves.

Next, observe that when $\sigma_{N_F}^2 \gg \sigma_{N_B}^2$, we have $\mathrm{I}(Y;B) > \mathrm{I}(Y;F)$ with strict equality (see Lemma 8).

From the definition of PID, $\mathrm{I}(Y;B) = \mathrm{Uni}(Y{:}B|F) + \mathrm{Red}(Y{:}B,F)$ and $\mathrm{I}(Y;F) = \mathrm{Uni}(Y{:}F|B) + \mathrm{Red}(Y{:}B,F)$.

Since $\mathrm{I}(Y;B) > \mathrm{I}(Y;F)$, we therefore have:

$$\mathrm{Uni}(Y{:}B|F) + \mathrm{Red}(Y{:}B,F) > \mathrm{Uni}(Y{:}F|B) + \mathrm{Red}(Y{:}B,F).$$

This leads to $\mathrm{Uni}(Y{:}B|F) > \mathrm{Uni}(Y{:}F|B) \geq 0$ since each PID term is nonnegative.

$\square$

