# OpenReview forum: "Formalizing Spuriousness of Biased Datasets using Partial Information Decomposition"
_ICLR.cc/2025/Conference — Submitted to ICLR 2025_

### Official Review · Reviewer_WNCj · 2024-10-29

**Soundness:** 3
**Presentation:** 3
**Contribution:** 3
**Rating:** 8
**Confidence:** 2

**Summary:**

This paper introduces a novel framework called spurious disentangler that uses Partial Information Decomposition (PID) to analyze and quantify spurious correlations in datasets. The authors propose a new measure $M_{sp}$ that assesses how likely a dataset will lead models to rely on spurious features over core features, implementing this through a three-module system of segmentation, dimensionality reduction, and PID estimation. Through experiments on multiple datasets, they demonstrate a consistent negative correlation between their spuriousness measure and model generalization metrics.

**Strengths:**

1. This paper clearly explains why simpler measures are insufficient and develops a novel spuriousness measure through examples and counterexamples.

2. This paper proposes a novel and complete framework called spuriousness disentangler for handling high-dimensional image data.

3. This paper provides extensive experimental results. It tests on multiple benchmark datasets, examines different types of sampling biases, and provides Grad-CAM visualizations. The experimental results well support their claims.

4. I think this research is of great significance. It can help identify problematic datasets before expensive model training.

**Weaknesses:**

1. This works requires manual identification of core and spurious features, which significantly limits its applicability since this might requires human-expert knowledge.

2. This paper focuses on the image classification task, it might be better if the authors can validate their framework on some NLP tasks.

**Questions:**

N/A

---

> ### Author Response · Authors · 2024-11-25
> **Author Response**
>
> We thank the reviewer for the review and appreciate the positive feedback on our work. We have made several edits to the main paper (marked in blue), including reorganizing the Main Results, including a new Fig. 3 to highlight the nuanced insights from PID, and performing additional experiments for automatic segmentation (Appendix A), tabular datasets (Appendix B), and other findings for the previous experiments (Appendix C.1).
>
> **(W1) On manual identification of core and spurious features**
>
> We acknowledge the reviewer’s concern regarding the potential limitation of manual identification of core and spurious features. In fact, there are extremely limited works that attempt to automatically separate core and spurious features without any other information. However, we have now incorporated automatic segmentation for separating the core and spurious features (see **Appendix A**), which does not rely on pre-defined training classes or explicit labels for spurious features or manual feature identification. The automatic segmentation utilizes open-vocabulary semantic segmentation model CLIPSeg[1] and performs segmentation of several objects. Then, one can at least choose the most relevant objects as “core,” e.g., “bird” object in Waterbirds using partial knowledge about the supervised classification goals. The remaining regions of the image can be chosen as “spurious.” To further demonstrate the practical applicability of our framework, we extend our framework for explainability on a tabular dataset where one might be interested in understanding and interpreting the nuanced statistical dependencies of a specific feature, e.g., “gender” with respect to other features (see **Appendix B**).
>
> **(W2) On validating the framework on NLP tasks**
>
> We appreciate the reviewer’s suggestion to explore NLP tasks for validating our framework. While this paper focuses primarily on image classification, the underlying principles of our approach can be extended to other domains, including NLP.  We have extended our framework to tabular dataset (see Appendix B). Investigating its application in NLP tasks would be an interesting direction for future work to further demonstrate the generalizability of our framework.
>
> [Reference]
>
> [1] Timo Lüddecke and Alexander Ecker. Image segmentation using text and image prompts. In
> Proceedings of the IEEE/CVF conference on computer vision and pattern recognition, pp. 7086–
> 7096, 2022.

---

> > ### Comment · Reviewer_WNCj · 2024-11-27
> >
> > Thanks for your response. I'd like to maintain my score. However, I want to emphasize that I'm not familiar with related works. Only as a non-expert, I think this paper is clearly written and its proposed method makes sense intuitively.

---

> > > ### Author Response · Authors · 2024-11-27
> > >
> > > Thank you so much for your positive opinion of our work. We are committed to making any other edits or answering any questions.

---

### Official Review · Reviewer_qqLo · 2024-11-03

**Soundness:** 2
**Presentation:** 3
**Contribution:** 1
**Rating:** 5
**Confidence:** 4

**Summary:**

This work focuses on the problem of spurious correlation in the data-driven models. It leverages the partial information decomposition (PID) to decompose the total information into four quantities such as the unique information of core and spurious features, the redundant information that is shared by two features, and the synergistic information that arises due to the collaboration of the two features. Based on this decomposition, the authors propose a framework called Spurious Disentangler to empirically evaluate the “spuriousness” of image data.

**Strengths:**

1. The paper focuses on an important task that evaluates the degree of “spuriousness” of a dataset.
2. The idea of decomposing the total information into the aforementioned four values to study the spurious correlation problem is very interesting.
3. The paper is well-written and easy to follow.

**Weaknesses:**

1. My main concern is the contribution and practicality of the proposed method. Although the idea of using information theory to quantify spuriousness is interesting, the actual use case of the proposed framework is limited and not properly discussed. Most results only show that the framework “is consistent with existing knowledge” (e.g. Theorem 1, experimental observations).

2. The proposed framework, “spuriousness disentangler”, relies heavily on segmentations. This greatly reduces its application scenarios. Datasets where spurious and core features can be explicitly separated as object & background are limited.

3. The requirement of the existence of a pre-trained semantic segmentation model is problematic. This is equivalent to requiring a much larger and more general dataset or a much more powerful model where the spurious correlation problems are already mitigated to a good extent. Such a “Deus Ex Machina” approach is questionable in practice.

4. The experiment section lacks insights and does not highlight the contribution of the proposed method.
    - The experiments are repeated on four datasets. However, the observations are all descriptive yet the contribution and the superiority of the proposed method are limited. For example, in L427-L429, the authors conclude from Fig. 7 that $M_{sp}$ is a good measure because it is consistent with worst-group accuracy. This claim treats worst-group accuracy as the standard for evaluating “spuriousness”. The contribution of PID is completely missing here.
    - The qualitative visualization of Figure 8 is only one sample. In L430, it is concluded from Figure 8 that “when the dataset is balanced or mixed background, the model emphasizes \textbf{more} on the core features (the red regions)”. This justification is insufficient. To justify that the model focuses “more” on the core feature. A score such as the IoU should be computed over the entire dataset to support this claim.


[Minor]:

1. L76, L78: “We first” appears twice.
2. L82-L83: The meaning of $A$ is not specified in $\mathrm{Syn}(Y:A,B)$. Is it supposed to be $F$? The definitions of $A$ and $F$ overlap throughout the manuscript and create unnecessary difficulty for the audience. The authors may consider unifying them for a clearer presentation.
3. In L147-148, $\mathcal{X}$ isn’t defined above.
4. In Figure 5, the location of the text “Encoder”, and “Decoder” and the curly brackets are misplaced.
5. The spacing after Figure 8 is completely missing.
6. It’s better to add captions for the subfigures in Figure 8 to indicate the five variants.

**Questions:**

1. Regarding the main concern, can the authors elaborate more on the use case of the framework where it is a better choice than existing measures such as worst-group accuracy?
2. In L53, the authors mentioned, “this notion of spuriousness in any given dataset has classically lacked a formal definition. To address this gap,…”. There are similar works discussing the quantification of spuriousness. e.g. [1,2]. Can the authors elaborate more on how the contribution of this work differs from this existing work?
3. Is it possible to generalize the proposed “spuriousness disentangler” framework to datasets where the segmentation of two features is infeasible? For example, in tabular data, the spurious features can be sensitive attributes such as gender, race, etc. These features cannot be segmented.
4. In counterexample 1, the authors refer to canonical example 1 and claim that this scenario should be considered as having “no spuriousness”. However, in canonical example 1, since $B = Y+N_B$, $F = Y+N_F$ with i.i.d. noise $N_B, N_F$, the spurious feature $B$ is equally connected to the label $Y$ compared with the core feature. Shouldn’t this be the “most spurious” scenario?


[Reference]

[1] Ye, H., Zou, J., & Zhang, L. (2023, April). Freeze then train: Towards provable representation learning under spurious correlations and feature noise. In *International Conference on Artificial Intelligence and Statistics* (pp. 8968-8990). PMLR.

[2] Wang, Y., & Wang, X. (2024, April). On the Effect of Key Factors in Spurious Correlation: A theoretical Perspective. In *International Conference on Artificial Intelligence and Statistics* (pp. 3745-3753). PMLR.

---

> ### Author Response · Authors · 2024-11-25
> **Author Response 1/2**
>
> We thank the reviewer for their review and comments. We have made several edits to the main paper (marked in blue), including reorganizing the Main Results, including a new Fig. 3 to highlight the nuanced insights from PID, and performing additional experiments for automatic segmentation (Appendix A), tabular datasets (Appendix B), and other findings for the previous experiments (Appendix C.1).  Below, we provide our responses to the comments, grouping some of them together.
>
> **(Q1 & W1-4) On the use-cases of the proposed method and relation to worst-group accuracy (WGA)**
>
> We would first like to clarify that our proposed technique is a dataset quality evaluation and explainability technique as opposed to model evaluation measures like worst-group accuracy (WGA) and SHAP/IoU. In essence, our goal is to develop explainability measures that are “pre-emptive” or “anticipative” of spuriousness using just the dataset before training the model since training a model can be expensive. The goal of our experiments is to show broad agreement between our anticipations from the dataset before training any model and the post-training behavior of actual models (when trained regularly to optimize performance without doing anything else specifically targeted towards avoiding spurious features).
>
> Our main **novelty** lies in distilling four types of statistical dependencies involving the core and spurious features, that goes beyond just saying whether a model relies on the spurious feature or not (**also see new Fig.3 in the updated paper**). To this end, we leverage the information-theoretic framework called Partial Information Decomposition (PID), with roots in statistical decision theory, which has not been studied in this context before. In particular, PID helps us formalize four types of scenarios as follows:
>
> $Uni(Y:F|B)>0$ captures scenarios where the feature $F$ is indispensable for prediction of $Y$ using Blackwell Sufficiency, and a Bayes optimal classifier will strongly rely on F.
>
> $Uni(Y:B|F)>0$ captures scenarios where the feature $B$ is indispensable for prediction of $Y$ using Blackwell Sufficiency, and a Bayes optimal classifier will strongly rely on B.
>
> $Red(Y:F,B)>0$ captures scenarios where either $F$ or $B$ suffice, and hence an optimal classifier could rely on either $F$ or $B$ or even both (the optimal classifier might not be an exclusive one, so it is ambiguous whether $F$ or $B$ is picked).
>
> $Syn(Y:F,B)>0$ captures an interesting scenario less explored in existing works. This is when both $F$ and $B$ are needed together by the optimal classifier, and neither of them suffice alone.
>
> We acknowledge that **identifying core and spurious features is challenging, and in fact, there is very limited work which attempts to address this problem automatically without using any other information**. Existing reweighting techniques [1] assume that the group information is available if not the exact pixels corresponding to the core feature. Some works [2] assume that an adversarially-robust model can sometimes identify the core features though with little interpretability. Since our work is focused on dataset evaluation/explainability prior to actual model training, we offer a workaround when the exact core/spurious features (pixels) are not known or even the group information is not available.
>
> APPENDIX A: ADDITIONAL EXPERIMENT 1: AUTOMATIC SEGMENTATION OF FEATURES
>
> Specifically, for supervised learning tasks, one might have some partial knowledge of what the core features should be even if we do not know the exact pixels, e.g., in Waterbirds, we know that the core feature is the “bird” object even if we do not know their exact pixels. In such scenarios, our proposition is to use automatic segmentation techniques (Open-Vocabulary Semantic Segmentation) such as CLIPseg [3] to at least perform object detection and then choose the most relevant objects as the core. Then, the regions of an image not associated with the “core” objects can often be considered spurious. Thus, our proposed technique can be applied in conjunction with such automatic segmentation techniques to interpret the nature of spuriousness and the anticipated behavior of the optimal predictor even without prior knowledge of the exact core and spurious feature split. See Appendix A for additional experiments on this setup.
>
> **Continued in next: TABULAR DATASETS**
>
> [Reference]
>
> [1] Shiori Sagawa, Pang Wei Koh, Tatsunori B Hashimoto, and Percy Liang. Distributionally robust neural networks for group shifts: On the importance of regularization for worst-case generalization. arXiv preprint arXiv:1911.08731, 2019.
>
> [2] Sahil Singla and Soheil Feizi. Salient imagenet: How to discover spurious features in deep learning? arXiv preprint arXiv:2110.04301, 2021.
>
> [3] Timo Lüddecke and Alexander Ecker. Image segmentation using text and image prompts. In Proceedings of the IEEE/CVF conference on computer vision and pattern recognition, 2022.

---

> ### Author Response · Authors · 2024-11-25
> **Author Response 2/2**
>
> APPENDIX B: ADDITIONAL EXPERIMENT 2: TABULAR DATASET (also Q3)
>
> To further demonstrate the practical applicability of our framework, we extend our framework to explainability on a tabular dataset where one might be interested in understanding and interpreting the nuanced statistical dependencies of a specific feature, e.g., “gender” with respect to other features. We studied the popular Adult dataset where the spurious feature is *gender*, and several other features are taken as the core. Dimensionality reduction is only performed on the core feature set because it is high-dimensional. Our technique is helpful in isolating biases in tabular datasets in this setting prior to training any model. Going beyond existing literature, we can pinpoint the nature of spurious dependencies and evaluate the quality of the dataset prior to model training.
>
> APPENDIX C.1:  ADDITIONAL RESULTS
>
> As per the reviewer’s comment, we have calculated the Intersection over Union (IoU) scores for the entire test dataset as well as for the minority test group of the Waterbird dataset (see Table 2).
>
> **(Q2) On discussion of suggested references**
>
> We include discussion on references [1] and [2] in the Related Works and can expand further if required.
>
> A recent work [2] looks into the problem of spurious correlations through the mathematical lens of separability of the spurious and core features under mixture of Gaussian assumptions. [2] also assume that the core and spurious feature split is given, and they define measures of separability to formalize the reliance on spurious feature by a model.
>
> Reference [1] (which was referenced before as well) looks into the problem of improving model training to reduce reliance on spurious features as opposed to pre-emptive dataset evaluation which is the focus of our work. We note that [1] also provides interesting insights on how the noise in the core feature plays a role in a model’s reliance on the core features.
>
> In contrast to these related works, our work isolates four specific types of inherent statistical dependencies in the dataset that provides a more nuanced understanding and dataset quality evaluation, going beyond solely identifying a model’s reliance on a specific feature. For instance, PID helps us isolate and understand four different types of statistical dependencies: when $F$ or $B$ is indispensable, when either $F$ or $B$ can be used, and when both $F$ and $B$ are required to be used jointly by the optimal classifier. PID provides a formal way of explaining when one feature is more informative than another, invoking Blackwell Sufficiency, which to the best of our knowledge, has not been explored in the context of understanding spurious correlations.
>
> **(Q4) On Redundant Information in Spuriousness**
>
> As we have mentioned, PID helps us isolate and understand four different types of statistical dependencies: when $F$ or $B$ is indispensable, when either $F$ or $B$ can be used, and when both $F$ and $B$ are required to be used jointly by the optimal classifier. *Specifically, redundant information captures inherent redundancies across features, i.e., ambiguous scenarios where an optimal predictor may pick either the core or spurious features without preference since either might suffice. This is different from the case where spurious feature $B$ is indispensable.* **Redundancy hints at the possibility that there are optimal predictors possible that would not rely on the spurious feature; hence it is not included in the spuriousness measure.**
>
> Nonetheless, should one wish to account for this type of statistical dependency as well, the measure of spuriousness can be modified to include this term as well.
>
> **On Minor Edits**
>
> We have incorporated these edits in the main paper.
>
> [Reference]
>
> [1] Ye, H., Zou, J., & Zhang, L. (2023, April). Freeze then train: Towards provable representation learning under spurious correlations and feature noise. In International Conference on Artificial Intelligence and Statistics (pp. 8968-8990). PMLR.
>
> [2] Wang, Y., & Wang, X. (2024, April). On the Effect of Key Factors in Spurious Correlation: A theoretical Perspective. In International Conference on Artificial Intelligence and Statistics (pp. 3745-3753). PMLR.

---

> > ### Comment · Reviewer_qqLo · 2024-12-02
> > **Thanks for the response**
> >
> > I have read through the rebuttal and appreciate the detailed response. I also appreciate that the revised manuscript has become more structured and easier to follow. The contributions of this paper are now properly highlighted. Overall, the idea and aim to identify patterns of spurious correlations from a data perspective is a very interesting topic. I have raised the score to 5 accordingly.
> >
> > However, some other important concerns persist:
> >
> > 1. Practicality: The four formalized types of biases are interesting. However, it remains unclear how this formulation can benefit the understanding of spurious correlation in practice, and how it can help in remedying the spurious correlation problem. As I stated in W1 of the original review, "Most results only show that the framework 'is consistent with existing knowledge' ". The question remains, "Why do we need this measure?".
> > 2. Validity: The goal of PID is "to develop explainability measures that are 'pre-emptive' or 'anticipative' of spuriousness using just the dataset". However, it has been observed in many previous works that different models and different training schemes (e.g. sample reweighting, regularizations) can lead to distinct performances on the **same** dataset. It remains unclear how the proposed PID can be justified as the intrinsic property of the dataset itself.

---

> > > ### Author Response · Authors · 2024-12-02
> > > **Response by Authors**
> > >
> > > We thank the reviewer for reviewing our paper again and for appreciating our contribution of evaluating spuriousness from a data perspective. Your response means a lot to us.
> > > Below, we provide our responses to these concerns:
> > >
> > > **Practicality and Validity:** Our main utility lies in improving dataset valuation, interpretability, and quality.
> > > We agree that there are several techniques for mitigating spurious correlations at the training stage on the **same** dataset, including reweighting or even adversarial training. Nonetheless, a measure fundamentally capturing and explaining spuriousness at its source (in the dataset itself) would be extremely useful as follows:
> > >
> > > 1. **Alternative to combating spurious correlations during training:** This will be useful if one does not have complete access to the entire training procedure, e.g., separate institutions/entities collect data and perform training. Our dataset explainability technique can provide “nutrition labels” [1] or “datasheets for datasets” [2] to inform the training process. Furthermore, adversarial training can sometimes be computationally expensive or require additional resources, motivating an alternate pathway of cleaning the dataset. *Drawing parallel from the field of algorithmic fairness, there are techniques to combat fairness in pre-processing (at the data level) and also in-processing (during training keeping the same dataset) which are both popular and applicable in different scenarios.*
> > >
> > > 2. **Intrinsic Property of the Dataset Itself:** We note that while the same dataset can lead to different models, the Bayes optimal classifier(s) are intrinsic to the dataset itself, and also the joint distributions of the random variables (input $X$ and label $Y$) on which our measures are calculated.
> > >
> > > 3. **Future work** would incorporate our measures for dataset selection or even as a regularizer for dataset cleansing, synthetic data generation, or data augmentation. Our technique of disentangling the nature of spuriousness in datasets could also have interesting connections with causal inference.
> > >
> > > 4. **Broader applicability in the era of pre-trained foundation models:** Having clean datasets for fine-tuning or evaluation is of utmost importance since we have limited control on the training process of large foundation models.
> > >
> > > Reference:
> > >
> > > [1] Yang, Ke, et al. "A nutritional label for rankings." Proceedings of the 2018 international conference on management of data. 2018.
> > >
> > > [2] Gebru, Timnit, et al. "Datasheets for datasets." Communications of the ACM 64.12 (2021): 86-92.

---

### Official Review · Reviewer_SCou · 2024-11-03

**Soundness:** 3
**Presentation:** 3
**Contribution:** 3
**Rating:** 6
**Confidence:** 1

**Summary:**

The framework builds on a foundation in information theory known as Partial Information Decomposition (PID) to break down the total information about the target variable into four distinct, non-negative components: unique information (within both core and spurious features), redundant information, and synergistic information. Using this decomposition, we introduce a novel metric for assessing the spuriousness of a dataset, guiding models to prioritize spurious features over core features.

**Strengths:**

this paper is based on a sound foundation: abstract / line 96 - line 125
the provided experimental evaluation doesn't include the statistical ratios (mean + std)

**Weaknesses:**

poor writing quality

**Questions:**

-

---

> ### Author Response · Authors · 2024-11-25
> **Author Response**
>
> We thank the reviewer for their review and appreciate the positive opinion about our work. We have now included the standard deviation for our results in Appendix C.3 Table 5. We have also included several new experiments in Appendix A, B, C and streamlined our writing in the Main Results section. We are also happy to incorporate any further suggestions that the reviewer might have.

---

### Official Review · Reviewer_pW3s · 2024-11-04

**Soundness:** 3
**Presentation:** 3
**Contribution:** 3
**Rating:** 5
**Confidence:** 2

**Summary:**

In this paper, the authors propose a novel measure of spuriousness by utilizing Partial Information Decomposition (PID) and an explainability framework consisting of segmentation, dimensionality reduction, and estimation modules to specifically handle high dimensional image data efficiently. In general, the proposed measure of spuriousness is interesting.

**Strengths:**

The proposed methods are novel and the experiments are extensive.

**Weaknesses:**

The writing in some places is a bit unclear and some implementation details are lacking.

**Questions:**

1, In the proposed autoencoder-based explainability framework, it seems that we need to select a non-negative constant $\gamma$ in dimensionality reduction phase, the readers may want to know how to select the value of $\gamma$. It will be helpful if the authors can give some guidance about the selection of $\gamma$.

2, In this paper, the authors propose a novel metric of spuriousness, then how can we identify one feature as a spurious one? It seems that we need the threshold?

3, It seems that there is a typo in "a the" in line 266.

4, In line 182, what is "Z_3 \bigoplus N$?

---

> ### Author Response · Authors · 2024-11-25
> **Author Response**
>
> We thank the reviewer for their positive opinion of our work. We have made several edits to the main paper (marked in blue), including reorganizing the Main Results, including a new Fig. 3 to highlight the nuanced insights from PID, and performing additional experiments for automatic segmentation (Appendix A), tabular datasets (Appendix B), and other findings for the previous experiments (Appendix C.1).
>
> **(Q1) On the choice of hyperparameter $\gamma$**
>
> We appreciate the reviewer’s suggestion. This constant $\gamma$, which lies in the range (0, 1), determines the weight given to the clustering loss and is treated as a hyperparameter. In our framework, we adopted the standard value of 0.1 for $\gamma$, which is commonly used in prior implementations of this clustering technique [1]. We have included this in Appendix C.3.1.
>
>
> **(Q2) On the threshold of $M_{sp}$ for identifying spurious feature**
>
> We appreciate the reviewer's comment. Our proposed measure of spuriousness $M_{sp}$ is designed to quantify the relative degree of spuriousness within a dataset. It is not intended to provide an absolute threshold for identifying individual features as spurious. However, by comparing $M_{sp}$ across different variants of a dataset, we can determine which dataset variant exhibits a higher or lower degree of spuriousness. In summary, a higher $M_{sp}$ ​ value indicates a more spurious dataset as compared to another prior to model training.
>
> **(Q3) On a typo in line 266**
>
> The typo has been corrected in the main paper.
>
> **(Q4) On $Z_3 \bigoplus N$**
>
> $\bigoplus $ represents modulo-2 addition, which corresponds to the elementwise exclusive OR (XOR) operation on binary values. In the specific example discussed in the paper, $Z_3$ and $N$ are Bernoulli random variables that independently take values of 0 or 1, each with a probability of 0.5. The operation $Z_3 \bigoplus N$ thus performs an elementwise XOR on these random variables.
>
> [Reference]
>
> [1] Xifeng Guo, Xinwang Liu, En Zhu, and Jianping Yin. Deep clustering with convolutional autoencoders. In Neural Information Processing: 24th International Conference, ICONIP 2017,
> Guangzhou, China, November 14-18, 2017, Proceedings, Part II 24, pp. 373–382. Springer, 2017.

---

### Official Review · Reviewer_2NSi · 2024-11-04

**Soundness:** 2
**Presentation:** 2
**Contribution:** 1
**Rating:** 3
**Confidence:** 4

**Summary:**

This paper presents a novel framework to quantify dataset spuriousness, addressing a gap in formalizing how spurious correlations between non-causal features and the label affect model generalization. The measure is calculated based on unique information and synergistic information values obtained from partial information decomposition. Experiments show negative correlation between the values of this measure and generalization metrics under distribution shift.

**Strengths:**

1)The problem is important and relevant to OOD Generalization.

2)The proposed measure is novel.

3)The experiments consider a range of datasets and somewhat empirically support the claims of the paper.

**Weaknesses:**

1)The measure relies on the assumption that causal and spurious features can be separated in the image as foreground and background. However, this assumption may not hold universally or even in most of the cases; for instance, spurious features like rotation or color affect all pixels rather than specific regions. In fact, disentangling causal and spurious features in a major challenge for many OOD tasks.

2)In the experiments, standard deviations or error bars are not provided, making it difficult to assess the scientific significance of the results.

3)There is no theoretical proof for why a higher value of the proposed measure would correspond to worse OOD performance.

4)Related to above, this paper lacks novel theoretical contribution. The theory presented is straightforward from partial information decomposition theory. Methodologically too, the main contribution comes from Bertschinger et al., (2014) which is used to calculate PID.

**Questions:**

1)During dimensionality reduction, how are the number of clusters chosen? Why do we need to approximate the distribution in a discrete way and what do we lose by doing so?

2)Why are other measures like I(Y;B) etc. not clearly reported in the results? This would help us compare the proposed measure with other measures.

---

> ### Author Response · Authors · 2024-11-25
> **Author Response 1/3**
>
> We thank the reviewer for their review. We have made several edits to the main paper (marked in blue), including reorganizing the Main Results, including a new Fig. 3 to highlight the nuanced insights from PID, and performing additional experiments for automatic segmentation (Appendix A), tabular datasets (Appendix B), and other findings for the previous experiments (Appendix C.1).
>
> **(W1) On the separation between core and spurious features**
>
> We acknowledge the reviewer’s concern regarding the assumption that core and spurious features can be separated as foreground and background. We agree that it is not always possible to separate the core and spurious features, particularly when spurious features, such as rotation or color, influence the entire image rather than specific regions. *In fact, there is extremely limited work that attempts to address this problem without any additional information.* We emphasize that our approach does not aim to explicitly label spurious features. Nonetheless, our strategy is applicable in several use-cases as a measure of dataset quality evaluation/explainability prior to expensive model training (without knowing the exact split of the core/spurious features apriori).
>
> *APPENDIX A: ADDITIONAL EXPERIMENT 1: AUTOMATIC SEGMENTATION OF FEATURES*
>
> In supervised classification tasks, we can approximately identify the core features by utilizing their association with the target label, e.g., “bird” object in Waterbirds dataset. To achieve this, we employ automatic segmentation using CLIPSeg [1], which isolates the foreground objects from the background without relying on pre-defined training classes or explicit labels for spurious features. We can choose the relevant objects as our core, e.g, “bird” object, and the remaining image as spurious. We can then compute the measure of spuriousness using this approximate split between core and spurious features and evaluate the quality of the dataset/interpret the nuanced dependencies using PID.
> | Waterbird Dataset | Red(Y:F,B) | Uni(Y:B\F)  | Uni(Y:F\B)  | Syn(Y:F,B) | $M_{sp}$ | Worst-group accuracy (\%) |
> |:-------------------|:------------|:----------|:----------|:----------|:----------|:---------------------------|
> | Unbalanced        | 0.00958    | 0.142746 | 1.09E-09 | 0.021737 | 0.12101  | 25.71                     |
> | Group Balanced    | 0.00433    | 0.00006  | 0.005405 | 0.022488 | -0.02783 | 85.82                     |
>
> Observe that there is a negative relationship between dataset spuriousness $M_{sp}$ and post-training metrics like the worst-group accuracy, as observed in the previous experiments. Hence, $M_{sp}$ quantifies spuriousness in this regard as well, thus providing pre-emptive insights into the dataset quality prior to training.
>
> *APPENDIX B: ADDITIONAL EXPERIMENT 2: TABULAR DATASET*
>
> To further demonstrate the practical applicability of our framework, we extend our framework for explainability on a tabular dataset where one might be interested in understanding and interpreting the nuanced statistical dependencies of a specific feature, e.g., “gender” with respect to other features. Specifically, we apply our method to the Adult [2] dataset, where the task is to predict whether an individual’s annual income exceeds 50k. We can choose a subset of features as the “core.” Since the core feature is high-dimensional, we employ k-means clustering to reduce dimensionality and discretize the features. Using our estimation module, we calculate PID values involving the core features, spurious features, and the target label (see Appendix B). For this set of experiments, the calculated $M_{sp}$ ​consistently exhibits a negative trend with worst-group accuracy, reinforcing the effectiveness of our approach.
> | Adult Dataset  | Red(Y:F,B) | Uni(Y:B\F) | Uni(Y:F\B) | Syn(Y:F,B) | $M_{sp}$ | Worst-group accuracy (\%) |
> |:----------------|:------------|:---------|:---------|:---------|:----------|:---------------------------|
> | Unbalanced     | 0.0374     | 0.0000  | 0.0661  | 0.0267  | -0.0928  | 10.63                     |
> | Group Balanced | 0.0000     | 0.0000  | 0.1163  | 0.0090  | -0.1252  | 63.96                   |
>
> In conclusion while disentangling causal and spurious features remains a significant challenge in OOD tasks, our method is still applicable to several use-cases for evaluating dataset quality without knowing the exact split of the core and spurious features apriori.
>
> [Reference]
>
> [1] Timo Lüddecke and Alexander Ecker. Image segmentation using text and image prompts. In
> Proceedings of the IEEE/CVF conference on computer vision and pattern recognition, pp. 7086–
> 7096, 2022.
>
> [2] Barry Becker and Ronny Kohavi. Adult. UCI Machine Learning Repository, 1996. DOI: https://doi.org/10.24432/C5XW20.

---

> > ### Author Response · Authors · 2024-11-25
> > **Author Response 2/3**
> >
> > **(W2) On bars for standard deviation**
> >
> > We have now included the standard deviations for our results in the Appendix C.3 Table 5.
> >
> > **(W3 and W4) On the novelty of contributions**
> >
> > We would like to highlight that the problem of *mathematically disentangling the nature of spurious correlations in the dataset* has received limited attention. Existing techniques such as data reweighting or alternate ways of model training have often shown empirical success in avoiding the spurious features; however, a fundamental understanding of the nature of spurious patterns in the dataset (in a “pre-emptive” or “anticipative” manner) prior to model training remains difficult.
> >
> > Our main **novelty** lies in isolating four different types of statistical dependencies involving the core and spurious features, bringing in the nuanced framework of Partial Information Decomposition (PID) that goes beyond simply saying whether a model is relying on spurious features or not (also **see new Fig. 3 in the main paper and other edits in the Main Results section**).
> > $Uni(Y:F|B)>0$ captures scenarios where the feature $F$ is indispensable for prediction of $Y$, and a Bayes optimal classifier will strongly rely on $F$.
> > $Uni(Y:B|F)>0$ captures scenarios where the feature $B$ is indispensable for prediction of $Y$, and a Bayes optimal classifier will strongly rely on $B$.
> > $Red(Y:F,B)>0$ captures scenarios where either $F$ or $B$ suffice, and hence an optimal classifier could rely on either $F$ or $B$ or even both (the optimal classifier is not an exclusive one, so it is ambiguous whether $F$ or $B$ is picked).
> > $Syn(Y:F,B)>0$ captures another interesting scenario less explored in existing works. This is when both $F$ and $B$ are needed together by the optimal classifier, and neither of them alone suffice.
> >
> > Then, our next contribution is to combine these PID terms into a novel measure of spuriousness that helps in dataset quality evaluation and explainability in a pre-emptive/anticipative manner prior to model training. We demonstrate that our “anticipative” measure of dataset spuriousness weakly correlates with other post-training measures of reliance on spurious features, such as Worst-Group Accuracy or SHAP perturbations and IoU.  To the best of our knowledge, this is the first work to leverage PID and Blackwell Sufficiency and bring in a non-trivial statistical decision-theory perspective into the problem space of spurious correlations, explicitly interpreting four nuanced dependencies. Following the reviewer’s comment, we have also reorganized our Main Results section to distill our findings into concrete lemmas.

---

> ### Author Response · Authors · 2024-11-25
> **Author Response 3/3**
>
> **(Q1) On clustering and discretization**
>
> We appreciate the reviewer for the relevant inquiries. The number of clusters is chosen as a hyperparameter through empirical validation. We tested different cluster numbers (e.g., 5, 10, 20) and analyzed the resulting PID values to observe variance which is low in our chosen range. The chosen number of clusters (10 in our case) ensures a balance between computational efficiency and loss of information. Excessively large clusters were avoided to prevent overfitting and also avoid high computational cost. The results are given below:
>
> | Unbalanced         | Red(Y:F,B) | Uni(Y:B\F) |  Uni(Y:F\B) | Syn(Y:F,B)   |
> |--------------------|------------|------------|-------------|--------------|
> | #Cluster 5         | 0.0065     | 0.1220     | 0.0000      | 0.0085       |
> | #Cluster 10        | 0.0057     | 0.1669     | 0.0000      | 0.0184       |
> | #Cluster 20        | 0.0025     | 0.1736     | 0.0000      | 0.0163       |
> | Standard Deviation | 0.0017     | 0.0229     | 0.0000      | 0.0043       |
> | Class Balanced     | Red(Y:F,B) | Uni(Y:B\F) |  Uni(Y:F\B) | Syn(Y:F,B)   |
> | #Cluster 5         | 0.0008     | 0.0221     | 0.0000      | 0.0012       |
> | #Cluster 10        | 0.0016     | 0.0300     | 0.0001      | 0.0114       |
> | #Cluster 20        | 0.0008     | 0.0128     | 0.0000      | 0.0097       |
> | Standard Deviation | 0.0004     | 0.0070     | 0.0000      | 0.0045       |
>
> Discretizing the distribution is necessary for numerical purposes because the current PID estimation tools and definitions are designed for discrete data. There is limited work on handling high-dimensional continuous data, e.g., [1] assumes Gaussian input random variables. Since our target variable is binomial, we cannot assume Gaussian distribution necessarily.
> Discretization causes some loss of information. To address this, we employed a self-learning approach using an autoencoder, which helps preserve as much information as possible during dimensionality reduction. This approach minimizes the impact of discretization while enabling us to leverage PID estimation effectively.
>
> **(Q2) On reporting $I(Y;B)$**
>
> We thank the reviewer for the suggestions. Notably, $I(Y;B)$ is the sum of $Uni(Y:B|F)$ and $Red(Y:F,B)$. We have now added the comparison of our proposed measure with this measure and other measures in Appendix C.1 Table 3.
>
> [Reference]
>
> [1] Praveen Venkatesh, Corbett Bennett, Sam Gale, Tamina Ramirez, Greggory Heller, Severine Durand, Shawn Olsen, and Stefan Mihalas. Gaussian partial information decomposition: Bias correction
> and application to high-dimensional data. Advances in Neural Information Processing Systems, 36,

---

> ### Author Response · Authors · 2024-11-27
> **Additional Clarification on Contributions**
>
> **Additional Clarification (Q2, W3 and W4)**
>
> Since our measure is a *pre-emptive* dataset evaluation measure prior to training, the **goal of our experiments** is to see how our anticipations from dataset agree with post-training model performance metrics like worst-group accuracy, SHAP, IoU, etc. **Notably, they are not mathematically the same thing – our measures anticipate spuriousness from dataset based on how the Bayes optimal classifier(s) *should* behave, and the empirical measures show how specific models actually behave (when trained regularly on that dataset without doing anything specific for spuriousness).**  Nonetheless, we observe an interesting correlation with our anticipative dataset quality measure and post-training model performance metrics on a broad range of experimental setups, further validating the efficacy of our approach.
>
> Also observe that for CelebA dataset, the *candidate measure $I(Y;B)$ does not exhibit a negative correlation with the worst-group accuracy.* However, our proposed measure aligns with this post-training trend, highlighting its effectiveness as an indicator for relative  dataset quality.
>
> | Dataset | Measures          | Unbalanced | Class Balanced | Group Balanced |
> |---------|-------------------|------------|----------------|----------------|
> | CelebA  | I(Y;B)            | 0.0238     | 0.0005         | 0.0151         |
> |         | Proposed $M_{sp}$ | -0.3091    | -0.3775        | -0.4797        |
> |         | W.G. Accuracy(\%) | 71.41      | 85.29          | 98.34          |

---

> ### Comment · Reviewer_2NSi · 2024-12-03
>
> I thank the authors for their response. However, several concerns still remain and I do not feel this paper is ready for publication at the moment. Hence, I cannot support acceptance.
>
> **W1** : I agree that the core and spurious features cannot be separated without additional information. However, this also means this measure is limited to only a few datasets and cannot be used broadly for all datasets.
>
> I do appreciate the tabular experiments by the authors though.
>
> **W2**: Thank you for adding the error bars. However, I would also expect error bars for Msp and other measures.
>
> **W3 + W4**: I respectfully disagree that the contribution is significant. Firstly, there are no theoretical guarantees that the proposed measure must negatively correlate with OOD Generalization. The paper proposes two candidate measures which are demonstrated to be unsuitable in some cases. But it not clear why this measure is suitable in all cases. Secondly, there is not a significant contribution in methodology. PID is calculated based on existing methods.
>
> **Q1**: Thank you for your answer. However, I suspect for high-dimensional data discretizing it would be a significant loss in information. This also relates to W4 since I believe this is a weakness in the methodology.
>
> **Q2**: Thank you. The numbers between the different measures do not seem to be very different in the empirical experiments and even I(Y;B) seems to negatively correlated with OOD Generalization. Also, is there any reason measures like I(Y;B|F) etc have not been considered?
>
>
>
> Also, I would recommend to improve the aesthetics of the plots in Figure 1 (although this does not affect my rating).

---

> > ### Comment · Reviewer_2NSi · 2024-12-03
> >
> > Further comments regarding **W4**,
> >
> > > analyzed the resulting PID values to observe variance which is low in our chosen range
> >
> > Correct me if I am wrong but this does not seem to be the case. The value of Red(Y:F,B) for Cluster # = 10 is double that of 5 and 20.
> >
> > Also, again for these results the error bars are not provided for each cluster. Since, an autoencoder is involved to discretize, there is a certain randomness. Hence, it is best to report error bars for all such results.

---

> > > ### Author Response · Authors · 2024-12-04
> > > **Response by Authors 1/2**
> > >
> > > We thank the reviewer for their review and for taking the time to share their thoughts and comments.
> > >
> > > **On Contributions and Broader Applicability:** We would like to emphasize that our primary contribution lies in identifying four types of statistical dependencies within datasets containing core ($F$) and spurious features ($B$), using the PID framework: when $F$ or $B$ is indispensable (unique information dominant), when either $F$ or $B$ suffice (redundant information dominant), and when both $F$ and $B$ are jointly needed (synergistic information dominant).  *This perspective is nontrivial and has not been explored in this context previously (that goes beyond whether a model is using or not using the spurious feature). In fact, we arrive at this from theoretical arguments on the anticipated behavior of the Bayes optimal classifier(s).* We have reorganized our Main Results and included a new Fig. 3. **Once the individual role of each PID term in this context is mathematically established, then we combine these PID terms to arrive at our measure adopting an axiomatic approach, studying examples and counterexamples.**
> > >
> > > The goal of our experiments is to show a correlation between our “preemptive” or “anticipative” measure of spuriousness on dataset, and post-training measures like worst-group accuracy, IoU, etc. **Notably, worst-group accuracy and our measure of spuriousness are not mathematically the same thing – our measures anticipate spuriousness from dataset based on how the Bayes optimal classifier(s) should behave and its feature preferences, and the empirical measures show how specific models actually behave (when trained on that dataset without doing anything specific for spuriousness).** Nonetheless, we observe an interesting correlation on a broad range of experimental setups, further validating the efficacy of our measure.
> > >
> > > Moreover, our framework is applicable, regardless of **whether the split for core and spurious features are provided as we show in Appendix A.** For instance, in image datasets, methods like CLIPSeg [1] can automatically segment objects and the framework is applicable to understand the nature of dependencies and spuriousness. Similarly, our framework can be extended to tabular data. Dataset valuation and interpretability are also of broader interest since one may not always have access to the training process of models, and combat spuriousness during training.
> > >
> > >
> > > **On $I(Y;B)$ and $I(Y;B|F)$:** We further substantiate our claims through canonical examples, where other measures fail to identify the dependencies between core and spurious features and the target label. For example, the measure I(Y;B) **does not** demonstrate a consistent negative correlation with out-of-distribution (OOD) generalization for CelebA dataset. In contrast, our proposed measure effectively captures these dependencies, underscoring its robustness and utility.
> > >
> > > The measure $I(Y;B|F)= Uni(Y:B|F)+Syn(Y:B|F)$ can also be computed using the PID terms and we will include it in our final version. A mathematical limitation of this measure is that it fails to distinguish between: (i) when $B$ is preferable over $F$ (unique information is dominant as in Lemma 3) and (ii) when both $B$ and $F$ are jointly needed (synergistic information is dominant as in Lemma 2). For the example in Lemma 2, where $ B=N$ and $F=Y+N$, both F and B can jointly give a much higher accuracy than either of them alone, and here $B$ might be denoising $F$. Also refer to the fourth case in Fig. 3.
> > >
> > > Reference:
> > >
> > > [1] Timo Lüddecke and Alexander Ecker. Image segmentation using text and image prompts. In
> > > Proceedings of the IEEE/CVF conference on computer vision and pattern recognition, pp. 7086–
> > > 7096, 2022.

---

> > > > ### Author Response · Authors · 2024-12-04
> > > > **Response by Authors 2/2**
> > > >
> > > > **On Dimensionality Reduction and Discretization:** The existing approaches to Partial Information Decomposition (PID) calculations struggle to handle continuous high-dimensional data effectively. There's [1] which only works for 1-D random variables and there's Gaussian PID [2] which assumes Gaussianity on the random variables. In this situation, dimensionality reduction and discretization is only natural (and also done in several prior works in information theory). Our self-learning step using an autoencoder prior to discretization (joint clustering and dimensionality reduction) is novel here as it attempts to capture the most significant patterns in the data, thereby minimizing the impact of discretization on the overall analysis. The clustering is not entirely lossless; the number of clusters is a hyperparameter that is chosen to balance between information loss and overfitting to the data. PID itself is also a relatively new and timely information-theory topic (compared to classical Shannon theory).
> > > >
> > > > References:
> > > >
> > > > [1] Ari Pakman, Amin Nejatbakhsh, Dar Gilboa, Abdullah Makkeh, Luca Mazzucato, Michael Wibral,
> > > > and Elad Schneidman. Estimating the unique information of continuous variables. Advances in
> > > > neural information processing systems, 34:20295–20307, 2021.
> > > >
> > > > [2] Praveen Venkatesh, Corbett Bennett, Sam Gale, Tamina Ramirez, Greggory Heller, Severine Durand, Shawn Olsen, and Stefan Mihalas. Gaussian partial information decomposition: Bias correction
> > > > and application to high-dimensional data. Advances in Neural Information Processing Systems, 36,
> > > > 2024.

---

### Author Response · Authors · 2024-11-27
**SUMMARY OF AUTHOR RESPONSE**

We thank the AC and reviewers for reviewing this paper. We appreciate that the reviewers have found our contributions novel and interesting. We have made several edits to the main paper (marked in blue), including reorganizing the Main Results, including a new Fig. 3 to highlight the nuanced insights from Partial Information Decomposition (PID), and performing **additional experiments for automatic segmentation (Appendix A), tabular datasets (Appendix B), and other findings for the previous experiments (Appendix C.1).**

**On Novelty:** We would like to reiterate that the problem of **mathematically explaining the nature of spurious correlations for dataset evaluation** has received limited attention. Existing techniques such as data reweighting or alternate ways of model training have often shown empirical success in avoiding the spurious features; however, *a fundamental understanding of the nature of spurious patterns in the dataset itself (in a “pre-emptive” or “anticipative” manner) prior to actual model training is less explored.*

Our main *novelty* lies in isolating four different types of statistical dependencies in a dataset involving the core ($F$) and spurious features ($B$), bringing in the nuanced framework of Partial Information Decomposition (PID) that has not been explored in this context before. We can identify when $F$ or $B$ is indispensable, or when either $F$ or $B$ can suffice (ambiguous), or when both are needed jointly, which goes beyond simply saying whether a model is relying on spurious features or not (also **see the new Fig. 3 in the main paper and other reorganizations of the Main Results into concrete lemmas**).

**On Separating Core and Spurious Features:**  There is limited work on automatically identifying core and spurious features without using any other information.  We now demonstrate applicability of our framework *even in scenarios where the core and spurious feature split is not known apriori.*

•	For instance, in **Appendix A**, we show how one can leverage automatic segmentation such as CLIPseg to at least identify various objects and then choose some of the most relevant objects as core, e.g., in supervised classification tasks on Waterbirds, one might know the “bird” object is core. The remaining image can be chosen as spurious for dataset quality evaluation.

•	In another use-case in **Appendix B**, we show how this framework can also be applied for explainability on tabular datasets, e.g., to understand the dependencies of one feature such as gender with respect to another subset of core features.

**Clarification on Experiments:** Since our measure is a *pre-emptive* dataset evaluation measure prior to training, the **goal of our experiments** is to see how our anticipations from dataset agree with post-training model performance metrics like worst-group accuracy, SHAP, IoU, etc. *Notably, they are not mathematically the same thing – our measures anticipate spuriousness from dataset based on how the Bayes optimal classifier(s) **should** behave, and the empirical measures show how specific models actually behave (when trained on that dataset without doing anything specific for spuriousness).*  Nonetheless, we observe an interesting correlation with our anticipative dataset quality measure and post-training model performance metrics on a broad range of experimental setups, further validating the efficacy of our approach.

We have also included the suggested references and are committed to making any other edits or answering any other questions. We look forward to the responses to our rebuttal.

---

### Meta-Review · Area_Chair_HTEr · 2024-12-17

**Metareview:**

This paper proposes a explanability framework to disentangle spuriousness by using partial information decomposition (PID). The paper's method can isolate four different types of statistical dependencies in a dataset involving the core and spurious features. The paper proposes a framework that has segmentation, dimensionality reduction, and estimation modules. Experiments show a tradeoff between the proposed measure of spuriousness and empirical model generalization metrics such as worst-group accuracy. The reviewers mention that the problem of spuriousness is an important topic, the proposed decomposition idea is interesting and novel, and experiments support the paper's claims.

There were two reviewers that gave high scores (6 and 8), but both of them privately communicated with the area chair that the area is outside their expertise and that their understanding of the paper is limited. I assigned less weight to the two compared to the other reviews.

The remaining three reviews gave lower scores (5, 5, 3). Reviewer pW3s mentioned writing is unclear and implementation details are missing. The authors provided further information. pW3s did not provide a final response but the rebuttal addresses the original concerns. Reviewer 2NSi mentioned a few weaknesses such as strong assumption about the spuriousness, lack of error bars in experiments, lack of theory of relationship between OOD performance, and lack of methodological novelty. The paper was updated with more experiments and error bars on some of the experiments. Although disagreements remained about the novelty, the discussions addressed the original concern to some extent and the novelty is clarified. Reviewer qqLo raised many important weaknesses. After some discussions, the reviewer raised the score from 3 to 5, but the reviewer's concerns regarding practicality and validity remained.

Based on the reviews and discussions and the summary above, I would like to recommend rejection this time, but hope the authors can further discuss practicality and validity, revise the paper, and resubmit in the future (A minor comment: I also recommend to include a Conclusion section. Without it, it would be logically strange and difficult for the reader to grasp the takeaways of this paper.)

**Additional Comments On Reviewer Discussion:**

See the meta-review above.

---

### Decision · Program_Chairs · 2025-01-22

Reject